# Unveiling the Power of Shared Spaces: A Gating-Driven Mechanism for Semi-Supervised Domain Adaptation

## Abstract

Domain adaptation (DA) aims to enhance the generalization ability of models in scenarios where labeled data in the target domain is scarce. In DA research, semi-supervised domain adaptation (SSDA) can utilize the labeled information in the target domain more effectively compared to unsupervised domain adaptation (UDA), thus achieving superior transfer performance and gaining widespread attention. Existing SSDA methods implicitly learn feature spaces in the process of aligning feature spaces between domains; however, the underlying mechanisms remain insufficiently explored. To address this issue, this paper first theoretically reveals the advantages of learning a shared feature space for enhancing transferability. Based on our theoretical insights, we develop a framework to learn a shared space, which is implemented by a gating-driven SSDA enhancement mechanism. It is feasible to explicitly filters out inconsistent features across domains compared with existing methods. Extensive experimental results demonstrate the significant improvements of the proposed gating-driven enhancement mechanism on state-of-the-art SSDA models. *Our code is anonymously provided in https://anonymous.4open.science/r/ICLR_8979.*

## 1 Introduction

Nowadays, deep learning has demonstrated significant effectiveness in various real-world tasks, including image recognition, segmentation, emotion analysis, and language translation (Akdemir & Barışçı, 2024; Dosovitskiy, 2020; Younesi et al., 2024). However, in practical applications, data labeling often faces challenges related to being expensive and time-consuming, resulting in large portions of data being unlabeled or sparsely labeled (Prabhu et al., 2021). To address this, knowledge from the annotated source domains can be transferred to unlabeled target domains to improve their performance, which is known as "domain adaptation" (DA) (Ganin & Lempitsky, 2015). DA methods can be mainly categorized into (1) unsupervised domain adaptation (UDA) (Long et al., 2015), which lacks labeled data in the target domain, and (2) semi-supervised domain adaptation (SSDA) (Saito et al., 2019), which has a small amount of labeled data in the target domain. Notably, compared to UDA, SSDA can leverage the labeled data to achieve better transfer performance, thus attracting widespread attention in recent researches (Saito et al., 2019; Ngo et al., 2024).

Existing deep SSDA primarily encompasses the following two categories of methods: adversarial-based and discrepancy-based methods (Farahani et al., 2021). Adversarial-based methods take adversarial mechanisms, such as generative adversarial networks (GANs) (Goodfellow et al., 2014), to reflect the different domains to similar feature space area (Li et al., 2021a; Saito et al., 2019). Discrepancy-based methods achieve alignment by focusing on minimizing the distributional discrepancy of features between two domains (Ngo et al., 2024; Saito et al., 2018), with the discrepancy

Table 1: Comparing existing methods of capturing feature views: MultiFea for capturing multi-perspective features, DataAug for augmenting data to increase generalization, and SharedFea for searching a shared feature space by alignment.

| Method | MultiFea | DataAug | SharedFea |
|---|---|---|---|
| MME (Saito et al., 2019) | × | × | ✓ |
| CDAC (Li et al., 2021a) | × | ✓ | ✓ |
| CLDA (Singh, 2021) | × | ✓ | ✓ |
| ECB (Ngo et al., 2024) | ✓ | ✓ | ✓ |
| LFL (Basak & Yin, 2024) | ✓ | × | ✓ |

including metrics such as maximum mean discrepancy (Long et al., 2015) and Wasserstein distance (Redko et al., 2017).

Clearly, these SSDA methods implicitly learn a **shared space** (Yousefnezhad et al., 2020; Basak & Yin, 2024) during the adaptation process due to the inclusion of a small amount of labeled data from the target domain, as shown in Table 1. Here, the shared space refers to a space that captures domain-invariant features such as the shape and sketch of objects, ensuring consistent feature distributions across domains (Yousefnezhad et al., 2020). Intuitively, by learning such a shared space, the classifier could exhibit more robust predictive performance on the features derived from both source and target domain data. However, there are still some critical research questions regarding the shared space that remain unanswered, such as:

- **RQ1:** For the models that address domain adaptation tasks, why is it essential to learn a shared feature space? What are the benefits of learning about shared spaces in solving SSDA problems?

- **RQ2:** What are the major challenges in learning an effective shared feature space for existing SSDA methods?

- **RQ3:** How can the above challenges be addressed to improve domain adaptation capabilities?

This paper aims to study the aforementioned questions, which reveal the mechanism of learning shared space, and further improve the performance of existing SSDA models.

First, we theoretically analyze performance guarantees for SSDA concerning the variation of the shared space (**RQ1**, w.r.t. Section 2.1 and 2.2). Unlike general shared-space studies (Basak & Yin, 2024; Yousefnezhad et al., 2020), we provide specific theoretical guarantees demonstrating that minimizing the number of domain-related features directly lowers the total variation distance and the target classification error bound. If the learned features are predominantly domain-related and fail to form a well-structured shared space, the total variation distance of feature distributions between two domains will be negatively affected. It will ultimately lead to a significant degradation in the model's DA performance. This offers a strict theoretical perspective on the advantages of learning in an effective shared space for SSDA.

Considering the benefits of shared space, we need to re-examine the existing SSDA methods and summarize the core challenge during learning shared space (**RQ2**, w.r.t. Section 2.3). Existing methods suffer from the problem that *features are extracted implicitly* (Basak & Yin, 2024), often retaining domain-specific styles during the alignment process due to insufficient training or model capacity. Accompanied by domain shifts and tasks for object recognition, it is hard for models to determine which features are exactly extracted during training. This issue becomes increasingly pronounced as domain discrepancy widens and the number of extracted features grows.

Based on theoretical insights and challenges of existing methods, we propose a conceptual framework to further explore the potential of the shared feature space, which can explicitly filter out domain-specific features. We introduce a gating-driven SSDA enhancement mechanism as feasible implementation of our framework (**RQ3**, w.r.t. Section 3). Specifically, by directly filtering out non-shared domain-related features through a gate network (Huang et al., 2020; Jiang et al., 2023), features in the shared space can be extracted more explicitly. It is worth noting that this framework is decoupled from the specific SSDA models, which means that the proposed enhancement mechanism can be seamlessly integrated into and further enhance the existing SSDA methods, which is flexible and scalable.

Our contributions can be concluded as follows:

- We reveal that existing approaches aim to learn a shared space and demonstrate the benefits of learning a shared space for SSDA by specific theoretical guarantees.

- Inspired by the theoretical insights, we propose a conceptual framework to explore the shared space that is decoupled from specific SSDA models, which explicitly filters out non-shared features through a gating mechanism, facilitating more effective learning in the shared space.

- Extensive experimental results show that the proposed enhancement mechanism helps the existing SSDA models select shared features more effectively and improve their domain generalization abilities significantly.

## 2 THEORETICAL ANALYSIS FOR SHARED SPACE

First, we define the necessary notations and representations of feature space in SSDA to bring the connection between shared space and SSDA, establishing a clear foundation for analysis. Then, we provide the theoretical analysis for shared space, which is based on the benefits and the challenges of learning shared space for existing SSDA models.

### 2.1 SHARED SPACE FOR SSDA

For source domain, we denote it by $D_S = \{(\mathbf{x}_{s_i}, y_{s_i})\}_{s_i=1}^{N_s}$. In target domain $D_T$, we denote the labeled data set as $D_{T_l} = \{(\mathbf{x}_{t_i}, y_{t_i})\}_{t_i=1}^{N_{tl}}$ and unlabeled data set as $D_{T_u} = \{(\mathbf{x}_{t_j})\}_{t_j=1}^{N_{tu}}$. In SSDA, the labeled data in target domain is very sparse, i.e., $N_{tl} \ll N_{tu}$. The purpose of SSDA is that train models on labeled dataset $D_l$, which includes $D_S$ and $D_{T_l}$, to achieve high performance on $D_{T_u}$.

Existing SSDA models (Saito et al., 2019; Li et al., 2021a; Singh, 2021) can be regarded as comprising two modules: a feature extractor $\mathcal{F}$ and a classifier $\mathcal{C}$. For any data $\mathbf{x}$, $\mathcal{F}$ extracts total feature $\mathbf{v}$ from it, i.e., $\mathbf{v} = \mathcal{F}(\mathbf{x})$. Then, classifier $\mathcal{C}$ predicts label on features, i.e., $\hat{y} = \mathcal{C}(\mathbf{v})$. Total feature $\mathbf{v}$ consists of $d$ features, i.e., $\mathbf{v} = [\mathbf{v}^1, \mathbf{v}^2, \cdots, \mathbf{v}^d] \in \Omega^d$ ($\Omega^d$ is the measurable set of all possible $\mathbf{v}$ with total $d$ subspace, and $\mathbf{v}^i \in \mathbb{R}^z$) [1]. For the classifier $\mathcal{C}$, $\mathbf{v}$ can be regarded as a feature sample of the feature variable $\mathbf{V}$.

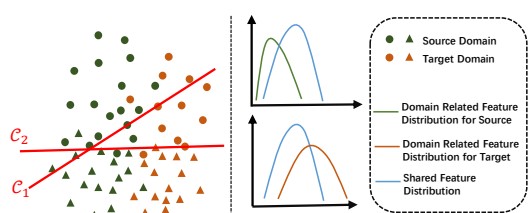

Figure 1: The feature distributions of different domains. The left part illustrates a 2-dimension feature space comprising 2 classes. The middle-top part of the figure represents distributions of the domain-related and shared features of the source domain. Middle-bottom part represents corresponding distributions for the target domain.

Specifically, the feature extractor updates and refines features based on feedback from the classifier's loss, moving toward the goal of learning a feature space that represents data from both domains. This space consists of two components (Basak & Yin, 2024): a **shared space** consists of shared features that are intrinsically relevant to the recognized objects, and a **domain-specific space** contains unique features that are relevant to the styles of the domain itself. Features in shared space are helpful to recognize object in classification for the learning model, such as sketch and shape of the object. The rest features in the domain-specific space mainly express the peculiarity of the picture rather than decide the boundary of the label function, such as style and background. As the examples shown in left part of Figure 1, the feature along the horizontal axis corresponds to domain-related feature, where the distribution between two domains exhibits significant discrepancies. It makes difficult for the classifier $\mathcal{C}_1$ to generalize directly to the target domain, which is trained on whole feature space of the source domain. In contrast, the feature along the vertical axis represents shared feature, whose distribution is consistent across domains, thereby enabling the classifier $\mathcal{C}_2$ to perform well on both domains, which is trained on shared feature. Therefore, the essence of the DA problem is to transfer features in the shared space while ignoring domain-related features.

Formally, for the source domain, we define the joint distribution of essential shared features in shared space as $\mathcal{P}_{ES}$ and the rest domain-related features in domain-specific space as $\mathcal{P}_{RS}$. In a slight abuse of notation, we define the distribution of shared features and the domain-related features as $\mathcal{P}_{ET}$ and $\mathcal{P}_{RT}$ for the target domain. With the feature number of each data being $d$, we set $d_r = \alpha d$ and $d_e = (1 - \alpha)d$, $\alpha \in [0, 1]$. $\alpha$ is the impact metric of shared space, which is more effective with it decreasing. The feature distribution of the entire source domain is produced by the combination of every feature distribution, i.e.,

$$\mathcal{P}_{D_S} = \mathcal{P}_{RS} \otimes \mathcal{P}_{ES} = \mathcal{P}_{rs}^1 \otimes \cdots \otimes \mathcal{P}_{rs}^{d_r} \otimes \mathcal{P}_{es}^1 \otimes \cdots \mathcal{P}_{es}^{d_e}. \tag{1}$$

Here, $\mathcal{P}_{rs}^i$ and $\mathcal{P}_{es}^j$ represent the $i$-th domain-related feature distribution and the $j$-th shared feature distribution in source domain, respectively. Also, they are probability density functions. Notice that

---

[1]Note that symbols without subscripts, such as $\mathbf{x}$ and $\mathbf{v}$, represent that they may come from any domain; otherwise we add subscripts $S$ ($T$) or $s$ ($t$) for them. For each data item $\mathbf{x}$, model $\mathcal{F}$ extracts total $d$ features and each feature is z-dimension.

samples $\mathbf{v}_s^i \sim \mathcal{P}_{rs}^i(\mathbf{V}^i)$ and $\mathbf{v}_s^j \sim \mathcal{P}_{es}^j(\mathbf{V}^j)$ are the $i$-th domain-related and $j$-th shared feature of $\mathbf{v}_s$, respectively. $\mathbf{V}$ represents the variable of the sample $\mathbf{v}$. $\otimes$ represents the product of distributions. Similarly, the entire target domain distribution is defined as

$$\mathcal{P}_{D_T} = \mathcal{P}_{RT} \otimes \mathcal{P}_{ET} = \mathcal{P}_{rt}^1 \otimes \cdots \otimes \mathcal{P}_{rt}^{d_r} \otimes \mathcal{P}_{et}^1 \otimes \cdots \mathcal{P}_{et}^{d_e}. \tag{2}$$

Based on the above perspective of measuring the entire domain distribution, **it is obvious that shared space is more dominated with decreasing** $\alpha$. The observation makes us ask: as an important role in affecting the quality of the alignment, how is the discrepancy between domains impacted by the variation of the learned shared space in theory ultimately?

## 2.2 BENEFITS OF LEARNING SHARED SPACE

In this part, we theoretically demonstrate that the error of SSDA is proportional to the distributional discrepancy, which is effectively mitigated by learning a well-structured shared space. We employ the total variation distance ($TV(\mathcal{P}_{D_S}, \mathcal{P}_{D_T})$) to quantify the distributional discrepancy between two domains, which is a common discrepancy measure (definition is shown in Appendix A).

**Error bound for SSDA**. We state the error bound of the target domain by binary classification problem. Assume the hypothesis function $h : \Omega^d \rightarrow \{0, 1\}$ for the data features $\mathbf{v}$, the error of $h$ for the source domain distribution can be defined as follows:

$$\epsilon_S(h) = \mathbb{E}_{\mathbf{v} \sim \mathcal{P}_{D_S}} |h(\mathbf{v}) - f_S^*(\mathbf{v})|, \tag{3}$$

here, $f_S^* : \Omega^d \rightarrow [0, 1]$ is the labeling function of source domain $D_S$, where the $f_S^*(\mathbf{v})$ represents the probability of label of $\mathbf{v}$ being 1. Also, $\epsilon_T(h)$ represents the error for the target domain $D_T$ regarding labeling function $f_T^*$. Following Ben-David et al. (2010), we can derive the theorem:

**Theorem 1.** *For any hypothesis $h \in \mathcal{H}$, where $\mathcal{H}$ is hypothesis space, it satisfies the following upper bound:*

$$\epsilon_T(h) \leq \epsilon_S(h) + TV(\mathcal{P}_{D_S}, \mathcal{P}_{D_T}) + \min\left\{\mathbb{E}_{D_T}\left[\left|f_T^*(\mathbf{v}) - f_S^*(\mathbf{v})\right|\right], \mathbb{E}_{D_S}\left[\left|f_S^*(\mathbf{v}) - f_T^*(\mathbf{v})\right|\right]\right\}. \tag{4}$$

Theorem 1 proves that reducing $TV(\mathcal{P}_{D_S}, \mathcal{P}_{D_T})$ effectively decreases the classification error $\epsilon_T(h)$ on the target domain for SSDA models. Given that the distributions of shared features are not the main contributors to domain discrepancy, we formulate the following reasonable assumption concerning TV:

**Assumption 1.** *The essential shared features and domain-specific features of two domains satisfy:*

- $TV(\mathcal{P}_{rs}^k, \mathcal{P}_{rt}^k) = \delta_k$, *for any* $k \in \{1, \cdots, d_r\}$.

- $TV(\mathcal{P}_{es}^k, \mathcal{P}_{et}^k) = 0$, *for any* $k \in \{1, \cdots, d_e\}$, *i.e.,* $\mathcal{P}_{es}^k = \mathcal{P}_{et}^k$.

We denote that the first $d_r$-th feature subspace of $\Omega^d$ is domain-specific space to facilitate understanding in the next analysis. To simplify the writing, the feature sampling value $\mathbf{v}_s^k$ in $\Omega_k$ is according to distribution $\mathcal{P}_{rs}^k$ ($\Omega_k$ is $k$-th feature subspace of $\Omega^d$). We can define feature sampling value $\mathbf{v}_t^k$ for target distribution in a similar manner.

Based on the above assumption, we discuss TV bounds of whole feature space from two cases: features are individual (Theorem 2) and non-individual (Theorem 3).

**i). Individual Features Case**. With features $\{\mathbf{v}^k\}_{k=1}^{d_r}$ being individuals for each other, we first present the TV bounds below.

**Theorem 2.** *Suppose that distributions of two domains satisfy the Assumption 1. For any $k \in \{1, \cdots, d_r\}$, we assume that a measurable subset $A_k \subset \Omega_k$, where the samples $\mathbf{v}_s^k \sim \mathcal{P}_{rs}^k$ and $\mathbf{v}_t^k \sim \mathcal{P}_{rt}^k$ satisfy $\mathbb{P}(\mathbf{v}_s^k \in A_k) - \mathbb{P}(\mathbf{v}_t^k \in A_k) = \delta_k$ and $\mathbb{P}(\mathbf{v}_t^k \in A_k) = \mu_k$. To simplify the writing, we set $\delta = \frac{1}{d_r}\Sigma_{k=1}^{d_r}\delta_k$. Then, $TV(\mathcal{P}_{D_S}, \mathcal{P}_{D_T})$ can be bounded as:*

$$TV(\mathcal{P}_{D_S}, \mathcal{P}_{D_T}) \geq 1 - 2\exp\frac{-\alpha d \delta^2}{2} \quad and \quad TV(\mathcal{P}_{D_S}, \mathcal{P}_{D_T}) \leq 1 - \prod_{k=1}^{\alpha d} \mu_k. \tag{5}$$

**Remark**. $\mu_k$ is a constant for each corresponding feature distribution and the product of them is increased with reducing $\alpha$. As defined in Section 2.1, $d_r = \alpha d$, $\alpha \in [0,1]$

**ii). Non-Individual Features Case**. Next, we extend our theorem into non-individual situation. If the features are not individual for each other, we can give a practical assumption that variable $\mathbf{V}^k$ is dependent on frontier variables by coefficient $\lambda_j$:

$$\mathbb{E}(\mathbf{V}^k|\mathbf{V}^{k-1} = \mathbf{v}^{k-1}, \cdots, \mathbf{V}^1 = \mathbf{v}^1) = \lambda_j \frac{\Sigma_{i=1}^{k-1}\mathbf{v}^i}{k-1} + (1-\lambda_j)\mathbb{E}(\mathbf{V}^k). \quad (6)$$

When $\lambda_j$ decreases to 0, the features are individual. The above assumption arises from a naturally occurring phenomenon and extends the applicability of previous theorem to non-individual case. For example, the contrast of an image is susceptible to be influenced by features such as color and brightness. The domain-related features can be split into $K$ independent subsets. Each subset is denoted as $N_j$, where $N_j$ concludes $n_j$ dependent features, $\Sigma_{j=1}^K n_j = \alpha d$. Independent subsets mean that for any $\mathbf{v}^{k_1} \in N_{j_1}, \mathbf{v}^{k_2} \in N_{j_2}(j_1 \neq j_2)$, $\mathbf{v}^{k_1}$ is independent with $\mathbf{v}^{k_2}$.

**Theorem 3.** *Suppose that distributions of two domains satisfy the Assumption 1 and conditions in Theorem 2 except for independence. Let samples of features $\{\mathbf{v}^k\}_{i=1}^{d_r}$ are sequentially drawn from $\mathbb{P}(\mathbf{V}^1, \cdots, \mathbf{V}^{d_r}) = \prod_{j=1}^K \mathbb{P}(N_j)$ and each sample satisfies Equation (6). $N_j$ is the independent subset which concludes $n_j$ dependent samples sequence $\{\mathbf{v}^1, \cdots, \mathbf{v}^{n_j}\}$ and $\mathbb{P}(N_j)$ is the joint distribution of $n_j$ features in subset $N_j$. For $\delta > \frac{\Sigma_{j=1}^K \lambda_j(n_j-1)}{d_r}$, the bounds of $TV(\mathcal{P}_{D_S}, \mathcal{P}_{D_T})$ are:*

$$TV(\mathcal{P}_{D_S}, \mathcal{P}_{D_T}) \geq 1 - 4\exp\frac{-2(\alpha d\delta/2 - \Sigma_{j=1}^K \lambda_j(n_j-1))^2}{\alpha d}, \quad (7)$$

*and*

$$TV(\mathcal{P}_{D_S}, \mathcal{P}_{D_T}) \leq 1 - \prod_{j=1}^K \mathbb{P}(\{\mathbf{v}_t^k\}^{n_j} \in \{A_k\}^{\otimes n_j}). \quad (8)$$

**Remark**. $\{A_k\}^{\otimes n_j}$ denotes the product of $A_k$ with size $n_j$, and $\{\mathbf{v}_t^k\}^{n_j}$ denotes the set of $\mathbf{v}_t^k$ with $n_j$ tuples, respecting to the target domain, where $k$ is feature index in subset $N_j$. $\mathbb{P}(\{\mathbf{v}_t^k\}^{n_j} \in \{A_k\}^{\otimes n_j})$ is increased with reducing $n_j$ and the product of them is increased with reducing $K$ in practice. Thus, the upper bound is decreased with reducing $\alpha$.

In conclusion, Theorems 2 & 3 demonstrate that a lower $TV(\mathcal{P}_{D_S}, \mathcal{P}_{D_T})$ is attributed to a smaller number of domain-related features (i.e., reducing $\alpha$), which focusing more on shared features. Specific proofs for above theorems are demonstrated in Appendix A.

## 2.3 CHALLENGES OF EXISTING MODELS

Along with learning the shared features, the existing SSDA models focus on the following objective functions to train:

$$\min \mathcal{L} = \mathcal{L}_l + \mathcal{L}_u + \mathcal{L}_{align}, \quad (9)$$

$\mathcal{L}_l$ is the based cross-entropy loss for labeled dataset $D_l$, and $\mathcal{L}_u$ is the loss for $D_{T_u}$, which can be the loss of assigning pseudo label for unlabeled data or augmentation data. $\mathcal{L}_{align}$ is considered to align features in domain-specific space. For adversarial-based methods, $\mathcal{L}_{align}$ can be formulated as entropy min-max process of unlabeled data to avoid models overfitting to source domain (Saito et al., 2019; Li et al., 2021a). For discrepancy-based, it could be the discrepancy of features respecting to cluster-level or instance-level of domains (Singh, 2021). These SSDA models' fundamental premise for achieving superior transfer performance based on the above loss function is their ability to extract a well-defined feature space during the training process.

However, most existing SSDA methods align features implicitly by optimizing objective functions hoping the model learns to suppress domain-specific features. This is a "black box" process. Due to factors such as the model's expressive capacity and insufficient training, **it is uncertain whether this implicit learning strategy can effectively extract features in the shared space**. That is, it may extract a large number of features highly specific to the domain. According to our theoretical results from Section 2.2, this will harm adaptation performance. Especially for CNN, it extracts the features by convolution, which concludes the pixel information in local windows (Krizhevsky et al., 2012). For the local field of vision in CNN, the domain-related and background information can be naturally absorbed and integrated into features.

## 3 Gating-driven Enhancement Mechanism

**High-level idea**. To address the above problems in existing models, an intuitive idea is to learn shared feature space more explicitly during training. When the domain-related features are explicitly filtered out in the models, the discrepancy of feature distributions is decreased, and the performance transfers better from the source domain to the target domain, which can be derived from the theoretical results.

To achieve the above purpose, we proposed a framework of learning shared space explicitly, which is implemented by the gating-driven mechanism. It takes advantage of the gate network (Huang et al., 2020; Jiang et al., 2023) to explicitly filter out some domain-related features and provide practical assistance to learn an effective, shared space. To ensure scalability, gate network is intentionally designed as a lightweight, channel-wise attention mechanism. The framework is shown in Figure 2. Overall, the gate network is positioned after $\mathcal{F}$, filtering the corresponding features explicitly. Then, the filtered features are fed into classifier $\mathcal{C}$ for further processing.

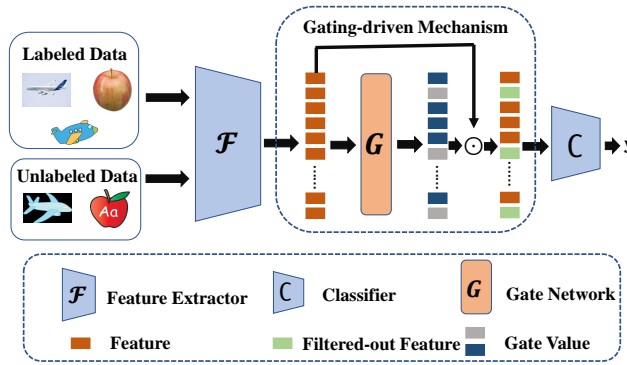

Figure 2: The framework of explicitly learning shared space by gating-driven mechanism. It is applied in the existing SSDA models. The gate network is placed in the position after $\mathcal{F}$ to output gate value for each feature and explicitly filter features. $\odot$ means the element-wise product.

Specifically, after inputting data $\mathbf{x}$ into feature extractor, the output of $\mathcal{F}$ is a concatenated feature vector $\mathcal{F}(\mathbf{x}) = [\mathbf{v}^1, \mathbf{v}^2, \cdots, \mathbf{v}^d]$. $d$ is the feature number, and $\mathbf{v}^i \in \mathbb{R}^z$ is the $i$-th feature where $z$ is the dimension of each feature. For every single feature $\mathbf{v}^i$, we compute the gate value by gate network to represent the importance of each feature for model classification:

$$g_i(\mathbf{v}^i) = \sigma(\mathbf{w}_i \cdot \mathbf{v}^i), \tag{10}$$

where $\mathbf{w}_i \in \mathbb{R}^{1*z}$ is the parameter vector (or value) of the linear layer for $i$-th feature and $\sigma$ is the activated function in gate network. In existing SSDA methods, the feature output of $\mathcal{F}$ has already undergone a flattening operation, which is a $d*1$ vector and each feature is a scalar in final, i.e., $z = 1$. For a feature vector $v \in \mathbb{R}^d$ (where $d = 512$ for ResNet34), the gate network consists of a single linear layer with parameters $w \in \mathbb{R}^d$ (one weight per channel), followed by a Sigmoid activation. This design adds only $d$ parameters, ensuring the method remains computationally efficient. To make full use of gating mechanisms, we combine the gate value $g_i(v^i)$ with the corresponding feature $\mathbf{v}^i$ to explicitly suppress domain-specific channels before the classifier:

$$\mathbf{e}_i(\mathbf{v}^i) = g_i(\mathbf{v}^i) \odot \mathbf{v}^i, \tag{11}$$

where $\odot$ means the element-wise product. Then, we aggregate total gated filtered features as gated feature embedding layer, i.e., $E_g(\mathbf{v}) = [\mathbf{e}_1(\mathbf{v}^1), \mathbf{e}_2(\mathbf{v}^2), \cdots, \mathbf{e}_d(\mathbf{v}^d)]$, which could select latent important information in the features. Then, we take $E_g(\mathbf{v})$ as the input of classifier $\mathcal{C}$. The total loss of gated feature embedding layer is summarized as:

$$\min \mathcal{L}(E_g(\mathbf{v})) = \mathcal{L}_l(\mathcal{C}(E_g(\mathbf{v}))) + \mathcal{L}_u(\mathcal{C}(E_g(\mathbf{v}))) + \mathcal{L}_{align}(E_g(\mathbf{v_s}), E_g(\mathbf{v_t})), \ \ s.t. \mathbf{v} = \mathcal{F}(\mathbf{x}), \tag{12}$$

here, $\mathbf{v_s}$ and $\mathbf{v_t}$ denote the features of source and target domain, respectively. The gate network optimizes the mechanism of filtering features according to the loss $\mathcal{L}(E_g(\mathbf{v}))$ from $\mathcal{C}$, which increases the impact of shared space.

**Relation with theoretical results**. From Theorems 2 & 3, as the number of domain-related features becomes smaller (i.e., reducing $\alpha$), $TV(\mathcal{P}_{D_S}, \mathcal{P}_{D_T})$ becomes lower and the influence of features in shared space gradually increasing. It finally reduces the classification error bound $\epsilon_T(h)$ and achieves better adaptation, as proved in Theorem 1. During training, the gate parameters $w_i$ are

Table 2: Accuracy (%) of SSDA methods under both 1-shot and 3-shot settings on DomainNet.

| Method | R→C | | R→P | | P→C | | C→S | | S→P | | R→S | | P→R | | Avg. | |
|---|---|---|---|---|---|---|---|---|---|---|---|---|---|---|---|---|
| | $1_{shot}$ | $3_{shot}$ | $1_{shot}$ | $3_{shot}$ | $1_{shot}$ | $3_{shot}$ | $1_{shot}$ | $3_{shot}$ | $1_{shot}$ | $3_{shot}$ | $1_{shot}$ | $3_{shot}$ | $1_{shot}$ | $3_{shot}$ | $1_{shot}$ | $3_{shot}$ |
| ENT (Grandvalet & Bengio, 2004) | 65.2 | 71.0 | 65.9 | 69.2 | 65.4 | 71.1 | 54.6 | 60.0 | 59.7 | 62.1 | 52.1 | 61.1 | 75.0 | 78.6 | 62.6 | 67.6 |
| DECOTA (Yang et al., 2021) | 79.1 | 80.4 | 74.9 | 75.2 | 76.9 | 78.7 | 65.1 | 68.6 | 72.0 | 72.7 | 69.7 | 71.9 | 79.6 | 81.5 | 73.9 | 75.6 |
| CLDA (Singh, 2021) | 76.1 | 77.7 | 75.1 | 75.7 | 71.0 | 76.4 | 63.7 | 69.7 | 70.2 | 73.7 | 67.1 | 71.1 | 80.1 | 82.9 | 71.9 | 75.3 |
| ProML (Huang et al., 2023) | 78.5 | 80.2 | 75.4 | 76.5 | 77.8 | 78.9 | 70.2 | 72.0 | 74.1 | 75.4 | 72.4 | 73.5 | 84.0 | 84.8 | 76.1 | 77.4 |
| G-ABC (Li et al., 2023) | 80.7 | 82.1 | 76.8 | 76.7 | 79.3 | 81.6 | 72.0 | 73.7 | 75.0 | 76.3 | 73.2 | 74.3 | 83.4 | 83.9 | 77.5 | 78.2 |
| EFTL (He et al., 2024) | 79.6 | 81.2 | 74.9 | 77.1 | 78.2 | 81.8 | 69.3 | 72.8 | 71.8 | 74.4 | 69.9 | 71.5 | 83.1 | 84.4 | 75.3 | 77.6 |
| IDMNE (Li et al., 2024) | 79.6 | 80.8 | 76.0 | 76.9 | 79.4 | 80.3 | 71.7 | 72.2 | 75.4 | 75.4 | 73.5 | 73.9 | 82.1 | 82.8 | 76.8 | 77.5 |
| LFL (Basak & Yin, 2024) | 80.9 | 81.1 | 79.9 | 80.2 | 80.1 | 81.1 | 73.7 | 76.8 | 79.2 | 82.5 | 78.4 | 78.5 | 86.9 | 90.1 | 78.7 | 81.2 |
| DARA (Wu et al., 2025) | 76.4 | 78.5 | 73.2 | 73.8 | 76.8 | 78.3 | 69.7 | 70.3 | 72.4 | 72.5 | 68.5 | 70.1 | 81.6 | 82.6 | 74.1 | 75.2 |
| MME (Saito et al., 2019) | 70.0 | 72.2 | 67.7 | 69.7 | 69.0 | 71.7 | 56.3 | 61.8 | 64.8 | 66.8 | 61.0 | 61.9 | 76.1 | 78.5 | 66.4 | 68.9 |
| **MME-G** | 72.0 | 73.9 | 69.8 | 71.4 | 70.4 | 73.0 | 61.5 | 63.7 | 66.6 | 68.8 | 64.0 | 65.1 | 78.3 | 80.1 | 68.9 | 70.9 |
| CDAC (Li et al., 2021a) | 77.4 | 79.6 | 74.2 | 75.1 | 75.5 | 79.3 | 67.6 | 69.9 | 71.0 | 73.4 | 69.2 | 72.5 | 80.4 | 81.9 | 73.6 | 76.0 |
| **CDAC-G** | 77.9 | 80.2 | 75.7 | 76.2 | 75.7 | 79.3 | 67.4 | 71.0 | 72.0 | 74.1 | 71.2 | 72.7 | 81.3 | 83.3 | 74.5 | 76.7 |
| ECB (Ngo et al., 2024) | 83.8 | **87.4** | 85.4 | 85.6 | 86.4 | 87.3 | 79.7 | 80.6 | 83.4 | 85.6 | 79.5 | 81.7 | 88.7 | 90.3 | 83.8 | 85.5 |
| **ECB-G** | **85.8** | 87.0 | **85.8** | **86.5** | **86.8** | **87.9** | **80.9** | **81.3** | **85.6** | **86.4** | **80.5** | **82.0** | **90.4** | **90.9** | **85.1** | **86.0** |

Table 3: Accuracy (%) of SSDA methods under 3-shot setting on Office-Home.

| Method | A→C | A→P | A→R | C→A | C→P | C→R | P→A | P→C | P→R | R→A | R→C | R→P | Avg. |
|---|---|---|---|---|---|---|---|---|---|---|---|---|---|
| ENT (Grandvalet & Bengio, 2004) | 61.3 | 79.5 | 79.1 | 64.7 | 79.1 | 76.4 | 63.9 | 60.5 | 79.9 | 70.2 | 62.6 | 85.7 | 71.9 |
| DECOTA (Yang et al., 2021) | 64.0 | 81.8 | 80.5 | 68.0 | 83.2 | 79.0 | 69.9 | 68.0 | 82.1 | 74.0 | 70.4 | 87.7 | 75.7 |
| CLDA (Singh, 2021) | 63.4 | 81.4 | 81.3 | 70.5 | 80.9 | 80.3 | 72.4 | 63.9 | 82.2 | 76.7 | 66.0 | 87.6 | 75.5 |
| ProML (Huang et al., 2023) | 67.8 | 83.9 | 82.2 | 72.1 | 84.1 | 82.3 | 72.5 | 68.9 | 83.8 | 75.8 | 71.0 | 88.6 | 77.8 |
| G-ABC (Li et al., 2023) | 67.3 | 83.8 | 80.4 | 69.2 | 83.9 | 80.2 | 70.5 | 69.3 | 82.8 | 76.0 | 70.0 | 88.1 | 77.2 |
| EFTL (He et al., 2024) | 70.3 | 84.8 | 83.8 | 70.6 | 84.6 | 81.5 | 72.6 | 70.9 | 85.4 | 77.5 | 72.8 | 89.3 | 78.7 |
| IDMNE (Li et al., 2024) | 66.4 | 82.4 | 79.3 | 69.1 | 83.1 | 79.5 | 69.0 | 67.6 | 82.7 | 75.2 | 71.7 | 88.1 | 76.2 |
| LFL (Basak & Yin, 2024) | 68.8 | 84.7 | 84.2 | 70.6 | 83.7 | 82.4 | 70.5 | 70.9 | 84.3 | 75.7 | 71.1 | 88.5 | 77.9 |
| DARA (Wu et al., 2025) | 70.9 | 87.8 | 72.9 | 82.1 | 17.6 | 69.2 | 82.8 | 69.8 | 81.0 | 79.4 | 68.5 | 83.0 | 76.5 |
| MME (Saito et al., 2019) | 63.6 | 79.0 | 79.7 | 67.2 | 79.3 | 76.6 | 65.5 | 64.6 | 80.1 | 71.3 | 64.6 | 85.5 | 73.1 |
| **MME-G** | 64.2 | 79.3 | 79.6 | 67.5 | 79.6 | 78.0 | 67.3 | 64.8 | 81.0 | 72.0 | 66.1 | 86.3 | 73.8 |
| CDAC (Li et al., 2021a) | 65.9 | 80.3 | 80.6 | 67.4 | 81.4 | 80.2 | 67.5 | 67.0 | 81.9 | 72.2 | 67.8 | 85.6 | 74.8 |
| **CDAC-G** | 65.9 | 81.6 | 80.4 | 67.8 | 81.3 | 80.0 | 68.1 | 67.3 | 82.1 | 73.2 | 68.3 | 86.0 | 75.2 |
| ECB (Ngo et al., 2024) | **78.7** | 90.2 | **91.3** | 85.2 | 90.4 | 91.0 | 83.9 | 76.8 | 91.2 | 85.6 | 77.6 | 92.8 | 86.2 |
| **ECB-G** | 78.6 | **91.6** | 91.1 | **86.4** | **91.6** | **91.8** | **85.1** | **78.5** | **91.8** | **87.3** | **79.6** | **93.1** | **87.2** |

updated via backpropagation from the classification loss $\mathcal{L}$. Since domain-specific features do not correlate with class labels across domains, the classifier naturally forces the gate to assign them lower weights ($g_i \to 0$) to minimize loss. While the mask is "soft" and learned via task loss, this structural intervention forces the model to make a distinct decision about feature importance for every channel, rather than relying solely on the "black box" weights of the backbone to handle domain shifts. This transforms the "implicit" alignment of previous methods into an "explicit" selection process, effectively increasing the dominance of shared features (reducing $\alpha$) and maximizing performance in the target domain . The results of $TV(\mathcal{P}_{D_S}, \mathcal{P}_{D_T})$, as shown in Table 5, also demonstrate that our proposed mechanism learns more on shared features with lower discrepancy.

**Difference with existing shared space learning**. Although some works (Basak & Yin, 2024; Yousefnezhad et al., 2020) attempt to address the challenges of learning shared spaces during DA, they typically follow a two-step learning process: first learning specific features tailored to each specific domain and then integrating these features across all domains. Bousmalis et al. (2016) constructs a shared feature space by employing a shared encoder alongside two private encoders, while Zhong et al. (2024) first learns an approximate shared space and subsequently fine-tunes it on the target domain. In contrast, our method explicitly filters features during the learning process, eliminating the need for further steps of adjustments. Although the mechanism seems simple, it is a theoretically inspired and appropriate choice. This simplicity incurs minimal computational cost (Appendix C.5), enabling efficient feature extraction and seamless integration with general SSDA models, offering high scalability.

## 4 EXPERIMENTS

### 4.1 SETUP

**Datasets**. We conducted experiments on 4 datasets, DomainNet (Peng et al., 2019), Office-Home (Venkateswara et al., 2017), Office-31 (Saenko et al., 2010) and VisDA-17 (Peng et al., 2018). Following the setup in Saito et al. (2019), we utilized 7 scenarios involving 4 domains in DomainNet, containing 140,006 images with 126 classes: Clipart (C), Sketch (S), Painting (P), and Real (R). Office-Home comprises 4 distinct fields: Art (A), Clipart (C), Product (P), and Real (R), and it includes 15,500 images with 65 classes. Office-31 contains 4,110 images with 31 classes across 3 domains: Amazon (A), Webcam (W) and DSLR (D). VisDA-17 is also a large-scale dataset with 12 categories, which includes 152,397 source synthetic images from 3D models and 55,388 real target images from real world. For fair comparisons, we selected 1 or 3 samples per class from target domain to assign labels and incorporated them into training process, following Saito et al. (2019). Also, above adaptation scenarios for each dataset are followed the standard protocols established in Saito et al. (2019); Li et al. (2021a); Ngo et al. (2024). This ensures our results are directly comparable to the vast majority of SSDA literatures which utilizes this exact setup.

**Implementation details**. The proposed gate network is adaptable to existing SSDA models. We combined our gate network with 3 currently popular models: **MME** (Saito et al., 2019), **CDAC** (Li et al., 2021a), and **ECB** (Ngo et al., 2024), which were denoted as **MME-G**, **CDAC-G** and **ECB-G**, respectively. The hyperparameters in our experiments were configured based on the recommendations from their works. More implementation details are provided in Appendix C.1.

**Baselines**. Except for above methods, we also compare with several state-of-the-art works: **ENT** (Grandvalet & Bengio, 2004), **CLDA** (Singh, 2021), **DECOTA** (Yang et al., 2021), **ProML** (Huang et al., 2023), **G-ABC** (Li et al., 2023), **LFL** (Basak & Yin, 2024), **DARA**(Wu et al., 2025), **EFTL**(He et al., 2024), **IDMNE** (Li et al., 2024) . We introduce them more specifically in Appendix C.2.

### 4.2 COMPARED RESULTS WITH STATE-OF-THE-ARTS

**Results for DomainNet**. In Table 2, we show the results of our algorithm on DomainNet, including 1-shot and 3-shot settings. Models integrated with the gate network demonstrate performance improvements in most scenarios compared to their original versions, with an average performance gain ranging from 0.9% to 2.5% for 1-shot and 0.5% to 2.0% for 3-shot. Notably, ECB-G achieves the highest average performance and consistently delivers optimal results in the majority of cases.

**Results for Office-Home**. We present the results on the Office-Home dataset under 3-shot setting in Table 3. ECB-G achieves an average accuracy of 87.2%, surpassing all other models. Additionally, MME-G and CDAC-G outperform the original models, w.r.t. MME and CDAC, in most scenarios.

Table 4: Accuracy (%) of SSDA methods under both 1-shot and 3-shot settings on Visda-17.

| Method | $1_{shot}$ | $3_{shot}$ |
|---|---|---|
| S+T | 60.1 | 63.2 |
| ENT (Grandvalet & Bengio, 2004) | 61.8 | 73.7 |
| CLDA (Singh, 2021) | 73.7 | 79.2 |
| MME (Saito et al., 2019) | 73.1 | 76.5 |
| **MME-G** | 75.6 | 78.0 |
| CDAC (Li et al., 2021a) | 74.0 | 78.1 |
| **CDAC-G** | 76.4 | 79.8 |
| ECB (Ngo et al., 2024) | 75.9 | 85.0 |
| **ECB-G** | **83.5** | **87.4** |

While our proposed gating-driven mechanism improves the overall average performance significantly, there are slight performance fluctuations in specific transfer directions like A→C and A→R. As detailed in Section 3, our method utilizes a gate network to explicitly filter out some domain-related features. However, in specific scenarios like A→C or A→R, certain features that are technically "domain-specific" might coincidentally aid classification when the domain gap is smaller or possesses specific overlaps. The baselines implicitly retain these features, potentially benefiting from these incidental cues. In contrast, our gating mechanism rigorously filters them out to enforce a stricter shared space. While this leads to a slight drop in these specific cases, it prevents the model from relying on spurious correlations, leading to better robustness across harder transfer tasks (e.g., R→P, where ECB-G improves by +1.9%).

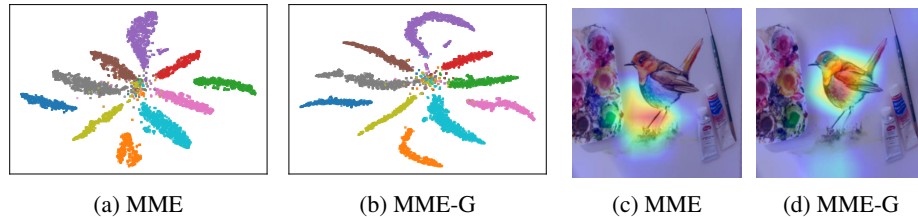

(a) MME    (b) MME-G    (c) MME    (d) MME-G

Figure 3: (a)-(b):t-SNE for our gate network with compared models. In each subfigure, we select samples from 10 classes, with each class represented by a unique color. Circles (o) indicate source domain data, while crosses (×) represent target domain data. The inter-class margins are distinct, and the overlap between source (o) and target (x) within clusters is tighter compared to 3(a). (c)-(d): Attention map by Grad-CAM for the "bird" class on DomainNet.

**Results for VisDA-17**. We present the accuracy results on the large-scale VisDA-17 dataset under 1-shot and 3-shot setting in Table 4. Clearly, ECB-G still achieves the best performance, reaching 83.5% and 87.4% in the 1-shot and 3-shot settings, respectively—surpassing the original ECB by 7.6% and 2.4%. Moreover, the gated-network-combined variants MME-G and CDAC-G also outperform their original counterparts (MME and CDAC), further confirming the effectiveness of the gating mechanism.

Results for Office-31 dataset and Office-Home under 1-shot setting are provided in Appendix C.3.

Since our gating mechanism is "decoupled from specific SSDA models", it serves as a complementary enhancement rather than just a competitor, consistently improving strong baselines like ECB.

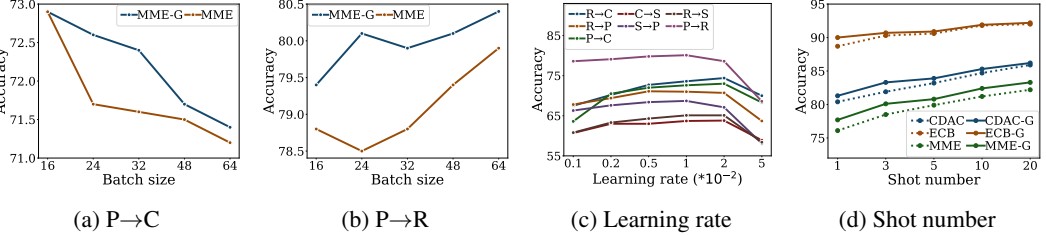

(a) P→C    (b) P→R    (c) Learning rate    (d) Shot number

Figure 4: (a)-(b): The results with different batch sizes for MME-G and MME under 3-shot setting on DomainNet. (c): The results with different learning rates for MME-G under 3-shot on DomainNet. (d): The results with different numbers of shot settings on P→R of DomainNet.

### 4.3 ANALYSIS

**Feature visualization**. To demonstrate the effectiveness of the proposed mechanism more intuitively, we use t-SNE (Van der Maaten & Hinton, 2008) to visualize the learned features on DomainNet transfer task P→R under the 3-shot setting. As shown in Figure 3a and 3b, the model integrated with the gate network produces more compact feature distributions for each category, with higher overlap between two domains, showing its better ability to learn shared space.

**Attention map visualization**. To verify whether the proposed gating-driven mechanism can better capture shared features, we utilize the Grad-CAM (Selvaraju et al., 2017) to visualize the attention maps of MME and MME-G, as shown in Figure 3c and 3d. Based on the Grad-CAM visualizations, it can be observed that MME-G focuses more on the "bird" object itself compared to the original MME, significantly reducing attention to background regions.

**Effectiveness of gating mechanism**. To further evaluate the effectiveness of proposed gating-driven mechanism, we also quantify the TV divergence between two domains with and w/o the gating layer on P→R of DomainNet, as shown in Table 5. With incorporation of the gating layer, the divergence $TV(\mathcal{P}_{D_S}, \mathcal{P}_{D_T})$ between two domains is significantly reduced, indicating that their feature representations in the projected space become more aligned and closer. Thus, as analyzed in Section 3, the results can further substantiate that our method effectively promotes the shared feature selection.

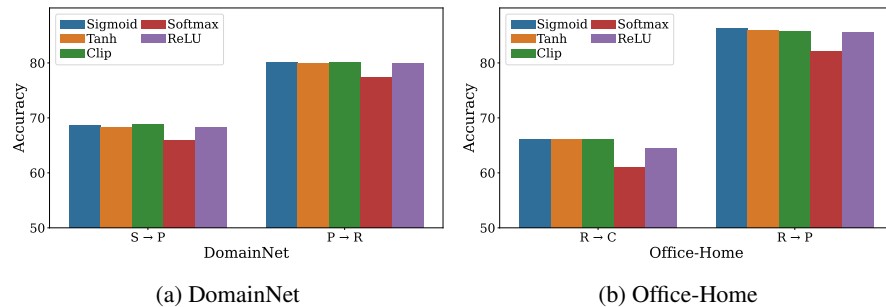

(a) DomainNet                (b) Office-Home

Figure 5: The results with different activation functions for MME-G under 3-shot on DomainNet and Office-Home.

**Effectiveness of varied batch sizes**. In Figure 4a and 4b, we present the performance for P→C and P→R under different batch sizes. For the P→C task, the prediction accuracy of both algorithms slightly decreases as the batch size increases. Meanwhile, for the P→R task, the performance generally improves with larger batch sizes. In general, across all batch sizes for both scenarios, our gating mechanism consistently enhances the adaptation performance of the model.

**Effectiveness of varied learning rates**. Figure 4c shows the performance of MME-G under different learning rates. As the learning rate increases, the prediction accuracy on the target domain initially improves, reaching its peak near 0.01, and then gradually decreases.

Table 5: TV value of features regarding to two domains.

| Method | MME | CDAC | ECB |
|---|---|---|---|
| w/o Gate | 0.108 | 0.106 | 0.155 |
| with Gate | 0.087 | 0.081 | 0.122 |

**Effectiveness of different numbers of shots**. We evaluate the performance with different numbers of shot settings in P→R of DomainNet. The number of selected labeled samples per class in target domain varies from 1 to 20. As shown in Figure 4d, the adaptation performance is gradually improved with increased numbers. Our gate network enhances model-predicted accuracy for each setting.

**Effectiveness of varied activation functions**. To evaluate the impact of different activation functions, we tested several options for the activation layer of the gate network, including Tanh, Softmax, ReLU, and Clip (direct clipping of gate values), as shown in Figure 5. Tanh and Clip achieve performance nearly on par with Sigmoid, while the other two activation functions under-perform. The primary reason is that Softmax introduces stronger interdependence among gate values for different features, while ReLU lacks an upper bound for gate values, leading to less effective gating. The performance differences among these five activation functions are similar on the DomainNet and Office-Home.

**More results**. We provide more results bout t-SNE, attention maps, multiple runs, effectiveness of activation functions and other parameters in the Supplement C.4 and C.5.

## 5 CONCLUSION

In this paper, we first theoretically analyze the benefits of learning shared space to SSDA. Based on the theory, we reveal the limitations of existing methods and propose a framework to better learn shared space for enhancing SSDA, which is implemented by gating-driven mechanism. Extensive experiments have proved the effectiveness of the proposed mechanism on state-of-the-art SSDA models. Beyond proposing the method, this work emphasizes the exploration of shared space, providing insights for the SSDA community. In the future, we plan to delve deeper into domain adaptation challenges from the perspective of shared feature space, exploring more sophisticated and effective mechanisms to further enhance SSDA.

ETHICS STATEMENT

This research was conducted independently, free from conflicts of interest or external sponsorship, The study adheres to ethical research principles, addressing considerations of discrimination, bias, fairness, privacy, security, and legal compliance while maintaining research integrity.

REPRODUCIBILITY STATEMENT

We have made every effort to ensure the reproducibility of our work in accordance with the ICLR reproducibility checklist. Specifically, we provide the source code in the abstract, and report detailed implementation parameters in the appendix. All datasets employed are well-established public benchmarks. For the theoretical contributions, we explicitly state the assumptions and present complete proofs in the appendix. Taken together, these materials enable independent verification and reproduction of our experimental and theoretical results.

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

APPENDIX

# A    PROOF FOR THEOREMS

**Definition 1.** *The total variation distance between probability density distributions $p$ and $q$ can be defined as follows:*

$$TV(p, q) = \sup_{A \in \mathcal{A}} |\mathbb{P}_p(A) - \mathbb{P}_q(A)|, \tag{13}$$

*where $\mathcal{A}$ is the collection of measurable subsets under $p$ and $q$. $\mathbb{P}_p(A)$ represents the probability measures of subset $A$ under $p$.*

## A.1    PROOF OF THEOREM 1

**Theorem** (1). *For any hypothesis $h \in \mathcal{H}$, where $\mathcal{H}$ is hypothesis space, it satisfies the following upper bound:*

$$\epsilon_T(h) \leq \epsilon_S(h) + TV(\mathcal{P}_{D_S}, \mathcal{P}_{D_T}) + \min \left\{ \mathbb{E}_{D_T}\left[|f_T^*(\mathbf{v}) - f_S^*(\mathbf{v})|\right], \mathbb{E}_{D_S}\left[|f_S^*(\mathbf{v}) - f_T^*(\mathbf{v})|\right] \right\}. \tag{14}$$

*Proof.*

$$\begin{aligned}
\epsilon_T(h) &= \epsilon_T(h) + \epsilon_T(h, f_S^*) - \epsilon_T(h, f_S^*) + \epsilon_S(h) - \epsilon_S(h) \\
&\leq \epsilon_S(h) + \left|\epsilon_T(h, f_T^*) - \epsilon_T(h, f_S^*)\right| \\
&\quad + \left|\epsilon_T(h, f_S^*) - \epsilon_S(h, f_S^*)\right| \\
&\leq \epsilon_S(h) + \mathbb{E}_{\mathbf{v} \sim D_T}\left[|f_T^*(\mathbf{v}) - f_S^*(\mathbf{v})|\right] \\
&\quad + \int |\mathcal{P}_{D_S} - \mathcal{P}_{D_T}| |h(\mathbf{v}) - f_S^*(\mathbf{v})| d\mathbf{v} \\
&\leq \epsilon_S(h) + \mathbb{E}_{D_T}\left[|f_T^*(\mathbf{v}) - f_S^*(\mathbf{v})|\right] + TV(\mathcal{P}_{D_S}, \mathcal{P}_{D_T})
\end{aligned} \tag{15}$$

If we take place of $\epsilon_T(h, f_S^*)$ by $\epsilon_S(h, f_T^*)$ in the first row of Equation (15), we will get the upper bound of $\epsilon_S(h) + \mathbb{E}_{D_S}\left[|f_S^*(\mathbf{v}) - f_T^*(\mathbf{v})|\right] + TV(\mathcal{P}_{D_S}, \mathcal{P}_{D_T})$ for the last row.

$\square$

## A.2    PROOF OF THEOREM 2

**Theorem** (2). *TV **bounds for features under individual case**. Suppose that distributions of two domains satisfy the Assumption 1. For any $k \in \{1, \cdots, d_r\}$, we assume that a measurable subset $A_k \subset \Omega_k$, where the samples $\mathbf{v}_s^k \sim \mathcal{P}_{rs}^k$ and $\mathbf{v}_t^k \sim \mathcal{P}_{rt}^k$ satisfy $\mathbb{P}(\mathbf{v}_s^k \in A_k) - \mathbb{P}(\mathbf{v}_t^k \in A_k) = \delta_k$ and $\mathbb{P}(\mathbf{v}_t^k \in A_k) = \mu_k$. To simplify the writing, we set $\delta = \frac{1}{d_r} \Sigma_{k=1}^{d_r} \delta_k$. Then, $TV(\mathcal{P}_{D_S}, \mathcal{P}_{D_T})$ can be bounded as:*

$$TV(\mathcal{P}_{D_S}, \mathcal{P}_{D_T}) \geq 1 - 2\exp\frac{-\alpha d \delta^2}{2} \quad and \quad TV(\mathcal{P}_{D_S}, \mathcal{P}_{D_T}) \leq 1 - \prod_{k=1}^{\alpha d} \mu_k. \tag{16}$$

*Proof.* According to the definition of total variation distance, we can obtain that: for any $k \in \{1, \cdots, d_r\}$, $\mathbb{P}(\mathbf{v}_t^k \in A_k) = \mu_k$, then $\mathbb{P}(\mathbf{v}_s^k \in A_k) = \mu_k + \delta_k$, where $\mathbb{P}(\cdot)$ represents the probability measures. For any sample $\mathbf{v}^k$, we use $I(\mathbf{v}^k) = 1$ to represent that sample $\mathbf{v}^k$ belongs to set $A_k$, otherwise $I(\mathbf{v}^k) = 0$.

According to Chernoff bound (Vadhan, 1999), we can get that:

$$\begin{cases} \mathbb{P}\left(\left(\Sigma_{k=1}^{d_r} I(\mathbf{v}_s^k) - (\mu + \delta)d_r\right) < -\frac{d_r \delta}{2}\right) < \exp\frac{-d_r \delta^2}{2}, \\ \mathbb{P}\left(\left(\Sigma_{k=1}^{d_r} I(\mathbf{v}_t^k) - \mu d_r\right) > \frac{d_r \delta}{2}\right) < \exp\frac{-d_r \delta^2}{2}. \end{cases} \tag{17}$$

Assume that set $A'$, consist of $d_r$ tuples, i.e., $\mathbf{v}^1, \cdots, \mathbf{v}^{d_r}$, contains at least $(\mu + \frac{\delta}{2})\alpha d$ samples that satisfy conditions in $\{\mathbf{v}^1 \in A_1, \cdots, \mathbf{v}^{d_r} \in A_{d_r}\}$. In other words, any $\{\mathbf{v}^k\}_{k=1}^{d_r} \in A'$, it satisfies $\Sigma_{k=1}^{d_r} I(\mathbf{v}^k) > (\mu + \frac{\delta}{2})\alpha d$. Thus, for any feature tuples $\{\mathbf{v}_s^k\}_{k=1}^{d_r}$ from source domain and feature

tuples $\{\mathbf{v}_t^k\}_{k=1}^{d_r}$ from target domain, both with $d_r$ tuples, we can bound total variation distance according to Chakraborty et al. (2024):

$$
\begin{aligned}
&TV(\mathcal{P}_{D_S}, \mathcal{P}_{D_T}) \\
=&TV(\mathcal{P}_{RS}, \mathcal{P}_{RT}) \\
\geq&\mathbb{P}(\{\mathbf{v}_s^k\}_{k=1}^{d_r} \in A') - \mathbb{P}(\{\mathbf{v}_t^k\}_{k=1}^{d_r} \in A') \\
=&\mathbb{P}\Big(\Sigma_{k=1}^{d_r} I(\mathbf{v}_s^k) > (\mu + \frac{\delta}{2})\alpha d_r\Big) \\
&- \mathbb{P}\Big(\Sigma_{k=1}^{d_r} I(\mathbf{v}_t^k) > (\mu + \frac{\delta}{2})\alpha d_r\Big) \\
\geq&(1 - \exp\frac{-d_r \delta^2}{2}) - \exp\frac{-d_r \delta^2}{2} \\
=&1 - 2\exp\frac{-\alpha d \delta^2}{2}
\end{aligned}
\tag{18}
$$

The first row of Equation (18) is shown according to that there is no discrepancy for distributions of shared features across domains in the Assumption 1.

For the set $A_k$, based on $\mathbb{P}(\mathbf{v}_s^k \in A_k) - \mathbb{P}(\mathbf{v}_t^k \in A_k) = \delta_k$ in Theorem 2 and $TV(\mathcal{P}_{rs}^k, \mathcal{P}_{rt}^k) = \delta_k$ in Assumption 1, we can infer that $\mathcal{P}_{rs}^k(\mathbf{v}^k) \geq \mathcal{P}_{rt}^k(\mathbf{v}^k)$ for any $\mathbf{v}^k \in A_k$. $\mathcal{P}_{rs}^k$ and $\mathcal{P}_{rt}^k$ are probability density functions. For any sample $\mathbf{v}^k$, we also use the $I(\mathbf{v}^k) = 1$ to represent that sampling value $\mathbf{v}^k$ belongs to set $A_k$, otherwise $I(\mathbf{v}^k) = 0$. Then, we have:

$$
\begin{cases}
\mathbb{P}\Big(\Sigma_{k=1}^{d_r} I(\mathbf{v}_s^k) = d_r\Big) = \prod_{k=1}^{d_r}(\mu_k + \delta_k), \\
\mathbb{P}\Big(\Sigma_{k=1}^{d_r} I(\mathbf{v}_t^k) = d_r\Big) = \prod_{k=1}^{d_r}\mu_k.
\end{cases}
\tag{19}
$$

Now we denote the set of $d_r$ tuples, $\{\mathbf{v}^1, \cdots, \mathbf{v}^{d_r}\}$, by $A''$ and every item of tuples in $A''$ satisfies $\{\mathbf{v}^1 \in A_1, \cdots, \mathbf{v}^{d_r} \in A_{d_r}\}$. It also means that $\Sigma_{k=1}^{d_r} I(\mathbf{v}^k) = d_r = \alpha d$. $\Omega^{d_r}$ is the measurable set of all possible $\{\mathbf{v}^k\}_{k=1}^{d_r}$, which can be regarded as whole space of $A''$, i.e, $A'' \subset \Omega^{d_r}$. Obviously, for any tuple $\{\mathbf{v}^k\}_{k=1}^{d_r}$ in $A''$, it satisfies $\mathcal{P}_{RS}(\{\mathbf{v}^k\}_{k=1}^{d_r}) > \mathcal{P}_{RT}(\{\mathbf{v}^k\}_{k=1}^{d_r})$. For the set of rest tuples in $\Omega^{d_r}$ satisfying the same condition, we define the set as $B''$, where $B'' \subset \Omega^{d_r} \backslash A''$. It can refer that, for any $\{\mathbf{v}^k\}_{k=1}^{d_r}$ which satisfies $\mathcal{P}_{RS}(\{\mathbf{v}^k\}_{k=1}^{d_r}) > \mathcal{P}_{RT}(\{\mathbf{v}^k\}_{k=1}^{d_r})$, it must belong to $B''$ or $A''$. Then we have:

$$
\begin{aligned}
&TV(\mathcal{P}_{D_S}, \mathcal{P}_{D_T}) \\
=&\Big(\mathbb{P}(\{\mathbf{v}_s^k\}_{k=1}^{d_r} \in A'') - \mathbb{P}(\{\mathbf{v}_t^k\}_{k=1}^{d_r} \in A'')\Big) \\
&+ \Big(\mathbb{P}(\{\mathbf{v}_s^k\}_{k=1}^{d_r} \in B'') - \mathbb{P}(\{\mathbf{v}_t^k\}_{k=1}^{d_r} \in B'')\Big) \\
\leq&\mathbb{P}(\{\mathbf{v}_s^k\}_{k=1}^{d_r} \in A'') - \mathbb{P}(\{\mathbf{v}_t^k\}_{k=1}^{d_r} \in A'') \\
&+ \mathbb{P}(\{\mathbf{v}_s^k\}_{k=1}^{d_r} \in \Omega^d \backslash A'') \\
=&\mathbb{P}\Big(\Sigma_{k=1}^{d_r} I(\mathbf{v}_s^k) = d_r\Big) - \mathbb{P}\Big(\Sigma_{k=1}^{d_r} I(\mathbf{v}_t^k) = d_r\Big) \\
&+ \Big(1 - \mathbb{P}\Big(\Sigma_{k=1}^{d_r} I(\mathbf{v}_s^k) = d_r\Big)\Big) \\
=&\prod_{k=1}^{d_r}(\mu_k + \delta_k) - \prod_{k=1}^{d_r}\mu_k + \Big(1 - \prod_{k=1}^{d_r}(\mu_k + \delta_k)\Big) \\
=&1 - \prod_{k=1}^{\alpha d}\mu_k
\end{aligned}
\tag{20}
$$

$\square$

## A.3 PROOF OF THEOREM 3

**Theorem** (3). *$TV$ **bounds for features under non-individual case**. Suppose that distributions of two domains satisfy the Assumption 1 and conditions in Theorem 2 except for independence. Let samples of features $\{\mathbf{v}^k\}_{i=1}^{d_r}$ are sequentially drawn from $\mathbb{P}(\mathbf{V}^1, \cdots, \mathbf{V}^{d_r}) = \prod_{j=1}^K \mathbb{P}(N_j)$ and each sample satisfies Equation (6). $N_j$ is the independent subset which concludes $n_j$ dependent samples sequence $\{\mathbf{v}^1, \cdots, \mathbf{v}^{n_j}\}$ and $\mathbb{P}(N_j)$ is the joint distribution of $n_j$ features in subset $N_j$. For $\delta > \frac{\Sigma_{j=1}^K \lambda_j(n_j-1)}{d_r}$, the bounds of $TV(\mathcal{P}_{D_S}, \mathcal{P}_{D_T})$ are:*

$$TV(\mathcal{P}_{D_S}, \mathcal{P}_{D_T}) \geq 1 - 4\exp\frac{-2(\alpha d\delta/2 - \Sigma_{j=1}^K \lambda_j(n_j-1))^2}{\alpha d}, \tag{21}$$

*and*

$$TV(\mathcal{P}_{D_S}, \mathcal{P}_{D_T}) \leq 1 - \prod_{j=1}^K \mathbb{P}(\{\mathbf{v}_t^k\}^{n_j} \in \{A_k\}^{\otimes n_j}). \tag{22}$$

*Proof.* For any $\mathbf{v}^k \in N_j$, it satisfies $\mathbb{E}(\mathbf{V}^k | \mathbf{v}^{k-1}, \cdots, \mathbf{v}^1) = \lambda_j \frac{\Sigma_{j=1}^{k-1}\mathbf{v}^j}{i-1} + (1-\lambda_j)\mathbb{E}(\mathbf{V}^k)$. we use the $I(\mathbf{v}^k) = 1$ to represent that feature samples $\mathbf{v}^k$ belongs to set $A_k$, otherwise $I(\mathbf{v}^k) = 0$. If $\delta > \frac{\Sigma_{j=1}^K \lambda_j(n_j-1)}{d_r}$, according to Chakraborty et al. (2024), it holds that :

$$\begin{cases} \mathbb{P}\Big( \big(\Sigma_{k=1}^{d_r} I(\mathbf{v}_s^k) - d_r(\mu + \delta)\big) > \frac{d_r\delta}{2} \Big) \\ \qquad\qquad < 2\exp\frac{-2(d_r\delta/2 - \Sigma_{j=1}^K \lambda_j(n_j-1))^2}{d_r}, \\ \mathbb{P}\Big( \big(\Sigma_{k=1}^{d_r} I(\mathbf{v}_t^k) - d_r\mu\big) > \frac{d_r\delta}{2} \Big) \\ \qquad\qquad < 2\exp\frac{-2(d_r\delta/2 - \Sigma_{j=1}^K \lambda_j(n_j-1))^2}{d_r}. \end{cases} \tag{23}$$

Also, we can denote the set of $d_r$ tuples by $A'$, i.e., $\{\mathbf{v}^1, \cdots, \mathbf{v}^{d_r}\}$, and $A'$ contains at least $(\mu + \frac{\delta}{2})\alpha d$ samples that satisfy $\mathbf{v}^k \in A_k$. Thus, for any feature set $\{\mathbf{v}_s^k\}_{k=1}^{d_r}$ of source domain and feature set $\{\mathbf{v}_t^k\}_{k=1}^{d_r}$ of target domain, both with $d_r$ tuples, it holds that:

$$\begin{aligned} TV&(\mathcal{P}_{D_S}, \mathcal{P}_{D_T}) \\ \geq& \mathbb{P}(\{\mathbf{v}_s^k\}_{k=1}^{d_r} \in A') - \mathbb{P}(\{\mathbf{v}_t^k\}_{k=1}^{d_r} \in A') \\ =& \mathbb{P}\Big( \Sigma_{k=1}^{d_r} I(\mathbf{v}_s^k) > (\mu + \frac{\delta}{2})\alpha d_r \Big) \\ &- \mathbb{P}\Big( \Sigma_{k=1}^{d_r} I(\mathbf{v}_t^k) > (\mu + \frac{\delta}{2})\alpha d_r \Big) \\ \geq& 1 - 4\exp\frac{-2(\alpha d\delta/2 - \Sigma_{j=1}^K \lambda_j(n_j-1))^2}{d_r} \end{aligned} \tag{24}$$

Due to the independence between subsets of $N_j$, we can replace the $\mathbb{P}(\mathbf{v}_t^k \in A_k) = \mu_k$ in Theorem 2 by $\mathbb{P}(\{\mathbf{v}^k\}^{n_j} \in \{A_k\}^{\otimes n_j})$. Also, we can define $A''$ and $B''$ in the similar way as in the proof of

Theorem 2, then the upper bound will be proved:

$$
\begin{aligned}
&TV(\mathcal{P}_{D_S}, \mathcal{P}_{D_T}) \\
&= \Big( \mathbb{P}\big(\{\{\mathbf{v}_s^k\}^{n_j}\}_{j=1}^K \in A''\big) - \mathbb{P}\big(\{\{\mathbf{v}_t^k\}^{n_j}\}_{j=1}^K \in A''\big) \Big) \\
&\quad + \Big( \mathbb{P}\big(\{\{\mathbf{v}_s^k\}^{n_j}\}_{j=1}^K \in B''\big) - \mathbb{P}\big(\{\{\mathbf{v}_t^k\}^{n_j}\}_{j=1}^K \in B''\big) \Big) \\
&\leq \mathbb{P}\big(\{\{\mathbf{v}_s^k\}^{n_j}\}_{j=1}^K \in A''\big) - \mathbb{P}\big(\{\{\mathbf{v}_t^k\}^{n_j}\}_{j=1}^K \in A''\big) \\
&\quad + \mathbb{P}\big(\{\{\mathbf{v}_s^k\}^{n_j}\}_{j=1}^K \in \Omega^{d_r}\backslash A''\big) \\
&= 1 - \prod_{j=1}^K \mathbb{P}\big(\{\mathbf{v}_t^k\}^{n_j} \in \{A_k\}^{\otimes n_j}\big)
\end{aligned}
\tag{25}
$$

$\square$

# B RELATED WORK

## B.1 DOMAIN ADAPTATION

Domain adaptation is crucial to address the problem of distribution shift between domains (Chen et al., 2019). Ganin & Lempitsky (2015); Long et al. (2018); Zhao et al. (2018) took advantage of adversarial learning to reflect the features of two domains into similar distributions. Long et al. (2015) aimed to reduce the discrepancy between domains by matching the mean embedding of domain distributions across multiple layers. Except for adversarial learning and reducing discrepancy for domain alignments, techniques such as entropy and pseudo labeling, which can extract valuable information from the target data, can also be applied to improve the model prediction for target domain (Pan et al., 2020; Vu et al., 2019). Li et al. (2021b) focused more on principal features and decreased the distribution discrepancy by semantic concentration. Xiao et al. (2023) utilized the graph spectral alignment to propagate neighborhood messages while considering more intra-domain information.

## B.2 SEMI-SUPERVISED DOMAIN ADAPTATION

With considering SSL more directly, Yang et al. (2021) separated the SSDA into two tasks, i.e., SSL task and UDA task, and leveraged the co-training framework to integrate the superiority of classifiers of both tasks. The co-training strategy was also adopted in Ngo et al. (2024), which took advantage of capturing global features of ViT (Dosovitskiy, 2020) and local features of CNN. In SSDA, adversarial training can enhance high-confident prediction for the target domain as well (Li et al., 2021a; Saito et al., 2019). To achieve both inter-domain and intra-domain adaptation, Huang et al. (2023); Singh (2021) maintained consistency of features in both domains from multiple views. Many of above methods assigned pseudo-labels to unlabeled data, while Yu & Lin (2023) focused on reassigning labels to the source domain data by pseudo center. Some works (Basak & Yin, 2024; Yousefnezhad et al., 2020) tried to learn shared spaces during DA, where they first learned specific features tailored to each specific domain, and then integrated these features across all domains.

## B.3 GATING MECHANISM

Due to the advantage of intensifying the important information in network layers, the gating mechanism is widely used in deep learning applications, especially for recommender systems (Geng et al., 2021; Ma et al., 2019). The hierarchical gate networks with feature-level and instance-level gate modules (Ma et al., 2019), effectively balanced long-term and short-term interests of users. The gating mechanism was applied to fuse features in multi-task and multi-domain recommendation (Chang et al., 2023). Also, for multi-task learning, Multi-gate Mixture-of-Experts Ma et al. (2018) utilized different gate networks to train each task. To achieve high click-through rate prediction, Huang et al. (2020); Jiang et al. (2023) adopted a gated structure to effectively choose feature information. Except for recommender system, the gating mechanism is also a common technique in computer

vision (Srivastava et al., 2015) and natural language process (Gehring et al., 2017), which is crucial for capturing long-term dependency.

## C  EXPERIMENTS

### C.1  IMPLEMENTATION DETAILS

The proposed gate network is adaptable to a variety of existing state-of-the-art SSDA models. Among them, we combined our gate network with three currently popular models: **MME** (Saito et al., 2019), **CDAC** (Li et al., 2021a), and **ECB** (Ngo et al., 2024). We denoted the three combined methods as **MME-G**, **CDAC-G** and **ECB-G**, respectively. To ensure fair comparisons, we kept the model architecture, initialization, optimizer, batch size and learning rate scheduler in our experiments consistent with their previous works. The hyperparameters in our experiments were configured based on the recommendations from their works. For DomainNet and Office-Home datasets, we chose the ResNet34 (He et al., 2016) as the backbone of $\mathcal{F}$. For the Office-31 dataset, we followed the recommendation in the previous papers (Saito et al., 2019; Ngo et al., 2024) and adopted AlexNet (Krizhevsky et al., 2012) as the backbone of $\mathcal{F}$. The activation function of our gate network was Sigmoid function. Similar to ECB, we additionally selected ViT (Dosovitskiy, 2020) as another backbone of feature extractor for ECB-G. All experiments were implemented by PyTorch and conducted on NVIDIA 4090 GPU.

### C.2  BASELINES

In this part, we introduce the baselines more specifically, which are compared in this work:

- **MME** (Saito et al., 2019) leveraged adversarial process on the entropy of unlabeled data to prevent the model from overfitting the source domain.

- **CDAC** (Li et al., 2021a) enhanced domain adaptation by incorporating data augmentation alongside its corresponding adversarial adaptive clustering loss.

- **ECB** (Ngo et al., 2024) captured both features from ViT and CNN and adopted a co-training strategy for them.

- **ENT** (Grandvalet & Bengio, 2004) is a based method that directly minimizes the entropy of the target domain, which encourages the model to produce confident and sharp predictions.

- **CLDA** (Singh, 2021) applied inter-domain and instance-level contrastive alignment to reduce inter-domain and intra-domain gaps, respectively.

- **DECOTA** (Yang et al., 2021) decomposed the SSDA into SSL and UDA and trained two classifiers for each task with the co-training framework.

- **ProML** (Huang et al., 2023) employed a prototype-based multi-level framework to learn the consistent features across different domains.

- **G-ABC** (Li et al., 2023) made use of adaptive betweenness clustering based on graphs to achieve semantic alignment for different domains.

- **EFTL** (He et al., 2024) proposed an effective target labeling framework which combine active learning and pesedo-label learning to select informative target data items.

- **IDMNE** (Li et al., 2024) generated new training samples by inter-domain mixup and leverage neighborhood expansion of target domain.

- **LFL** (learn, forget, and learn more) (Basak & Yin, 2024) utilized the strategies of "learn", "forget", and "learn more" to obtain domain-agnostic features, which is essential for adaptive classification tasks.

- **DARA** (Wu et al., 2025) aligned the representations from probability-level and feature-level to decrease the discrepancy of two domains.

## C.3 COMPARED RESULTS WITH STATE-OF-THE-ARTS

### C.3.1 RESULTS FOR OFFICE-HOME

We present the results on Office-Home under the 1-shot in Table 6. Due to the original paper (Li et al., 2023) only demonstrating results of Office-Home with 3-shot, we did not report the 1-shot results here. As shown in Table 6, ECB-G still achieves the best mean performance for the dataset. Although the performance in the 1-shot setting is slightly lower than in the 3-shot setting, models with gating-driven mechanisms outperform their previous corresponding versions in most scenarios, with the average accuracy gain ranging from 0.1% to 0.6%.

### C.3.2 RESULTS FOR OFFICE-31

We demonstrate the results on Office-31 dataset in Table 7. As shown in the table, our method consistently outperforms existing approaches even on such relatively small-scale datasets with limited image quantities and categories. The performance gain is particularly notable in the 1-shot setting, where our method consistently enhances the performance of baseline approaches, achieving gains of up to 2% in many cases. The comparative results reported here are directly taken from the corresponding original papers.

Table 6: Accuracy (%) of SSDA methods under 1-shot setting on Office-Home.

| Method | A→C | A→P | A→R | C→A | C→P | C→R | P→A | P→C | P→R | R→A | R→C | R→P | Mean |
|---|---|---|---|---|---|---|---|---|---|---|---|---|---|
| ENT (Grandvalet & Bengio, 2004) | 52.9 | 75.0 | 76.7 | 63.2 | 73.6 | 73.2 | 63.0 | 51.9 | 79.9 | 70.4 | 53.6 | 81.9 | 67.9 |
| DECOTA (Yang et al., 2021) | 42.1 | 68.5 | 72.6 | 60.3 | 70.4 | 70.7 | 60.0 | 48.8 | 76.9 | 71.3 | 56.0 | 79.4 | 64.8 |
| CLDA (Singh, 2021) | 56.3 | 76.1 | 79.3 | 66.3 | 73.9 | 76.3 | 66.2 | 55.9 | 81.0 | 72.6 | 60.2 | 83.2 | 70.6 |
| ProML (Huang et al., 2023) | 64.5 | 79.7 | 81.7 | 69.1 | 80.5 | 79.0 | 69.3 | 61.4 | 81.9 | 73.7 | 67.5 | 86.1 | 74.6 |
| EFTL (He et al., 2024) | 65.7 | 80.5 | 80.8 | 65.6 | 79.6 | 77.5 | 68.7 | 63.3 | 82.6 | 74.3 | 66.6 | 87.2 | 74.4 |
| LFL (Basak & Yin, 2024) | 64.1 | 80.1 | 81.1 | 70.6 | 79.5 | 79.1 | 67.9 | 62.5 | 80.9 | 75.2 | 69.1 | 87.9 | 74.8 |
| MME (Saito et al., 2019) | 59.6 | 75.5 | 77.8 | 65.7 | 74.5 | 74.8 | 64.7 | 57.4 | 79.2 | 71.2 | 61.9 | 82.8 | 70.4 |
| MME-G | 60.7 | 75.7 | 77.7 | 65.4 | 75.0 | 74.5 | 64.5 | 58.2 | 79.3 | 71.0 | 62.9 | 83.7 | 70.7 |
| CDAC (Li et al., 2021a) | 61.2 | 75.9 | 78.5 | 64.5 | 75.1 | 75.3 | 64.6 | 59.3 | 80.0 | 72.7 | 61.9 | 83.1 | 71.0 |
| CDAC-G | 61.3 | 78.0 | 79.1 | 65.3 | 75.1 | 75.4 | 62.9 | 58.7 | 79.5 | 71.7 | 63.3 | 83.4 | 71.1 |
| ECB (Ngo et al., 2024) | 72.9 | 88.3 | 89.6 | **84.8** | **91.3** | **89.5** | 82.9 | 71.2 | 89.9 | **85.5** | 75.4 | 92.0 | 84.4 |
| ECB-G | **74.6** | **89.4** | **89.8** | 84.7 | 89.9 | 89.2 | **85.0** | **73.1** | **90.5** | **85.5** | **76.5** | **92.3** | **85.0** |

Table 7: Accuracy (%) of SSDA methods under both 1-shot and 3-shot settings on Office-31.

| Method | W→A | | D→A | | Avg. | |
|---|---|---|---|---|---|---|
| | $1_{shot}$ | $3_{shot}$ | $1_{shot}$ | $3_{shot}$ | $1_{shot}$ | $3_{shot}$ |
| ENT (Grandvalet & Bengio, 2004) | 50.7 | 64.0 | 50.0 | 66.2 | 50.4 | 65.1 |
| CLDA (Yang et al., 2021) | 64.6 | 70.5 | 62.7 | 72.5 | 63.6 | 71.5 |
| G-ABC (Li et al., 2023) | 67.9 | 71.0 | 65.7 | 73.1 | 66.8 | 72.0 |
| DARA (Wu et al., 2025) | 66.1 | 71.8 | 65.7 | 72.0 | 65.9 | 71.9 |
| MME (Saito et al., 2019) | 57.2 | 67.3 | 55.8 | 67.8 | 56.5 | 67.6 |
| **MME-G** | 58.2 | 67.7 | 57.3 | 68.4 | 57.8 | 68.1 |
| CDAC (Li et al., 2021a) | 63.4 | 70.1 | 62.8 | 70.0 | 63.1 | 70.0 |
| **CDAC-G** | 65.9 | 70.4 | 64.5 | 70.6 | 65.2 | 70.5 |
| ECB (Ngo et al., 2024) | 77.9 | 85.2 | 76.3 | 84.0 | 77.1 | 84.6 |
| **ECB-G** | **80.7** | **86.7** | **79.0** | **84.5** | **79.9** | **85.6** |

## C.4 MORE VISUALIZATION RESULTS

### C.4.1 FEATURE VISUALIZATION

In Figure 6, we show the visualization of the feature space by t-SNE on P→R of DomainNet. The figure shows the visualization for CDAC-G and ECB-G, with their original models. Obviously,

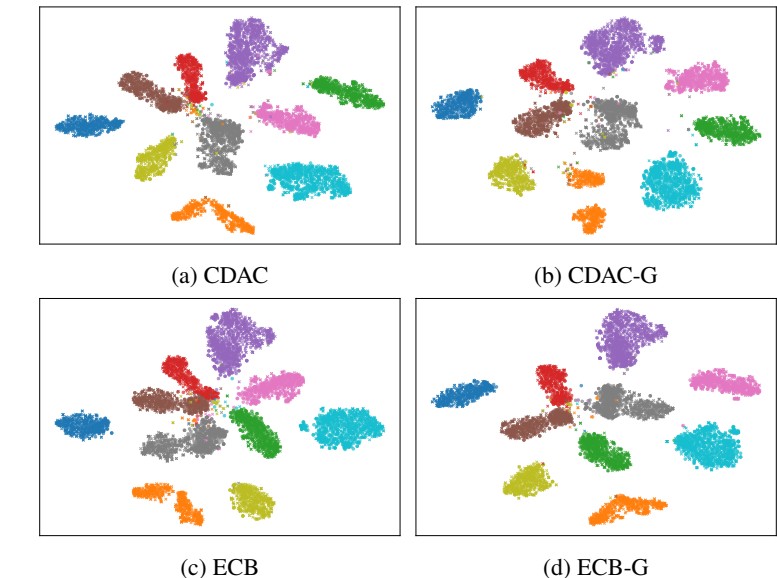

|     |     |
| --- | --- |
| (a) CDAC | (b) CDAC-G |
| (c) ECB | (d) ECB-G |

Figure 6: t-SNE for our gate network with compared models. In each subfigure, we select samples from 10 classes, with each class represented by a unique color. Circles (o) indicate source domain data, while crosses (×) represent target domain data.

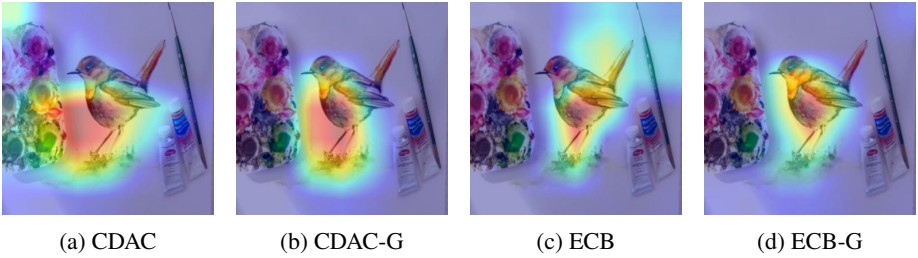

| (a) CDAC | (b) CDAC-G | (c) ECB | (d) ECB-G |

Figure 7: Attention map visualization by Grad-CAM for the "bird" class on DomainNet.

our gating-driven mechanism produces the features distributed more compactly and achieves better alignment.

### C.4.2 ATTENTION MAP VISUALIZATION

We show the attention maps by Grad-CAM to visualize the gradient attention of CDAC-G and ECB-G in Figure 7 for the "bird" class. Also, we provide attention map visualization for the "bus" class in Figure 8. It is evident that incorporating the gating mechanism enables the model to focus more effectively on the shared features that are intrinsically relevant to the recognized targets.

### C.5 MORE ANALYSIS RESULTS

#### C.5.1 EFFECTIVENESS OF VARIED BATCH SIZES

The results of performance on the rest tasks of DomainNet under different batch sizes are shown in Figure 9a to 9e. It is obvious that across almost all scenarios with different batch sizes, our gating-driven mechanism could improve the adaptation performance of the original models.

#### C.5.2 EFFECTIVENESS OF TIME COMPLEXITY

We provide the running time complexity of our methods in Table 8, including the seconds required for both training and inference. It is evident that our gating mechanism does not significantly in-

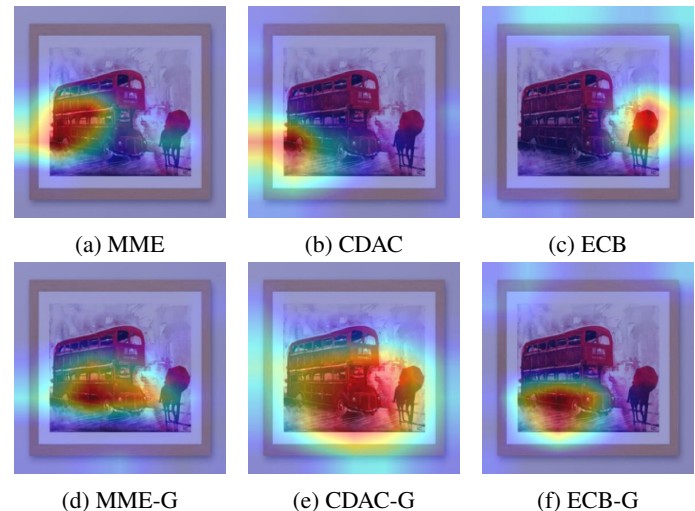

Figure 8: Attention map visualization by Grad-CAM for the "bus" class on DomainNet.

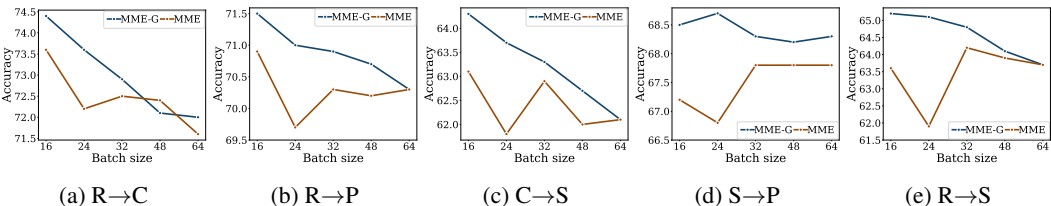

Figure 9: The results with different batch sizes for MME-G and MME under 3-shot setting on DomainNet.

crease the time complexity. In addition, we present a specific GFLOPs comparison in the Table 9 to validate efficiency of our gating-driven mechanism, demonstrating that it does not occupy large computation resource.

### C.5.3 RESULTS OF MULTIPLE RUNS

We report the t-test results over 3 runs of our gating-driven mechanism compared with original baselines without gating mechanism in Table 10. It is well established that a t-test value less than or equal to 0.05 indicates a statistically significant difference between two groups. As shown in most cases, the incorporation of our gating-driven mechanism leads to statistically significant improvements in average performance compared to the original SSDA methods.

### C.5.4 EFFECTIVENESS OF VARIANT GATED NETWORK DESIGN

We compare several gate network designs, including CNN, Transformer, Attention, and MLP. Table 11 and 12 report their performance on DomainNet under the 3-shot and 1-shot settings, respectively. As shown, the gate network used in our paper—based on a Sigmoid activation—achieves the best average performance among all compared designs. Since the backbone of our feature extractor is based on a ResNet architecture, using Attention or Transformer as the gate network leads to a performance drop compared to the other designs, likely due to architectural mismatch and suboptimal feature interaction in this setting.

Table 8: The time complexity of during training and inference on P→R of DomainNet (seconds).

| Method | MME | MME-G | CDAC | CDAC-G | ECB | ECB-G |
|---|---|---|---|---|---|---|
| Train | 20102 | 20466 | 28323 | 28395 | 66486 | 67688 |
| Inference | 97 | 113 | 117 | 116 | 94 | 95 |

Table 9: GFlops of SSDA methods with or without gate network.

| | MME | CDAC | ECB |
|---|---|---|---|
| w/o gate | 3.682266624 | 3.682266624 | 20.550929664 |
| with gate | 3.682269184 | 3.682269184 | 20.550932324 |

### C.6 MORE APPLICATIONS ABOUT GATING-DRIVEN MECHANISM

Although our theoretical framework regarding shared spaces (Section 2) is generalizable. this paper focuses on validating the gating mechanism for image-based SSDA. In future work, we plan to further investigate its applicability in UDA, multi-source domain adaptation, zero-shot domain shifts and non-visual domains, which are beyond the current focus of this paper. We believe the mechanism is applicable to UDA. To demonstrate this, we integrated our mechanism into the ECB framework (this SSDA method are also suitable to UDA) for a UDA setting. In Table 13, the preliminary results show that the gating mechanism successfully improves UDA performance, further validating that explicit feature filtering benefits adaptation even without target labels.

## D THE USE OF LARGE LANGUAGE MODELS (LLMs).

In this work, large language models (LLMs) were employed solely for the purpose of refining and improving the clarity of written expressions. No other uses of LLMs, such as retrieval, discovery, or research ideation, were involved in the preparation of this manuscript.

Table 10: t-test results of multiple runs for incorporating gating-driven mechanism compared with original baselines under both 1-shot and 3-shot settings on DomainNet.

| Method | R→C $1_{shot}$ | R→C $3_{shot}$ | R→P $1_{shot}$ | R→P $3_{shot}$ | P→C $1_{shot}$ | P→C $3_{shot}$ | C→S $1_{shot}$ | C→S $3_{shot}$ | S→P $1_{shot}$ | S→P $3_{shot}$ | R→S $1_{shot}$ | R→S $3_{shot}$ | P→R $1_{shot}$ | P→R $3_{shot}$ |
|---|---|---|---|---|---|---|---|---|---|---|---|---|---|---|
| **MME-G** | 0.0022 | 0.0066 | 0.0025 | 0.0102 | 0.0160 | 0.0128 | 0.0001 | 0.0013 | 0.0029 | 0.0003 | 0.0018 | 0.0007 | 0.0121 | 0.0005 |
| **CDAC-G** | 0.1567 | 0.0571 | 0.0041 | 0.0246 | 0.1835 | 0.2495 | 0.0350 | 0.1150 | 0.0202 | 0.0830 | 0.0058 | 0.0377 | 0.0160 | 0.0041 |
| **ECB-G** | 0.0051 | 0.0153 | 0.0098 | 0.0153 | 0.1835 | 0.0034 | 0.0462 | 0.0025 | 0.0551 | 0.1019 | 0.0890 | 0.0152 | 0.0397 | 0.0130 |

Table 11: Accuracy (%) of different gate network design on DomainNet under 3-shot.

| Method | R→C | R→P | P→C | C→S | S→P | R→S | P→R | Avg |
|---|---|---|---|---|---|---|---|---|
| MLP | 73.6 | 70.8 | 72.8 | **64.1** | 68.9 | 64.8 | 78.9 | 70.6 |
| Attention | 72.2 | 70.2 | 72.5 | 63.1 | 67.4 | 63.4 | 78.9 | 69.7 |
| Transformer | 73.4 | 70.9 | 72.7 | 63.1 | 68.5 | 64.6 | 79.2 | 70.3 |
| CNN | 73.7 | 70.8 | 72.9 | 63.9 | **69.2** | 64.7 | 79.8 | 70.7 |
| Sigmoid | **73.9** | **71.4** | **73.0** | 63.7 | 68.8 | **65.1** | **80.1** | 70.9 |

Table 12: Accuracy (%) of different gate network design on DomainNet under 1-shot

| Method | R→C | R→P | P→C | C→S | S→P | R→S | P→R | Avg |
|---|---|---|---|---|---|---|---|---|
| MLP | 71.6 | 69.3 | **70.4** | **62.2** | **66.7** | 63.5 | 77.9 | 68.8 |
| Attention | 70.0 | 68.6 | 69.7 | 60.4 | 65.6 | 61.1 | 77.6 | 67.6 |
| Transformer | 71.1 | 68.9 | 70.2 | 60.9 | 66.1 | 63.3 | 77.9 | 68.3 |
| CNN | 71.1 | 69.2 | 70.0 | 61.8 | 67.1 | 63.9 | 78.1 | 68.7 |
| Sigmoid | **72.0** | **69.8** | **70.4** | 61.5 | 66.6 | **64.0** | 78.3 | **68.9** |

Table 13: Accuracy (%) of UDA methods on Office-Home.

| Method | A→C | A→P | A→R | C→A | C→P | C→R | P→A | P→C | P→R | R→A | R→C | R→P | Avg |
|---|---|---|---|---|---|---|---|---|---|---|---|---|---|
| DANN (Ganin et al., 2016) | 45.6 | 59.3 | 70.1 | 47.0 | 58.5 | 60.9 | 46.1 | 43.7 | 68.5 | 63.2 | 51.8 | 76.8 | 57.6 |
| MCD (Saito et al., 2018) | 48.9 | 68.3 | 74.6 | 61.3 | 67.6 | 68.8 | 57.0 | 47.1 | 75.1 | 69.1 | 52.2 | 79.6 | 64.1 |
| BNM (Cui et al., 2020) | 52.3 | 73.9 | 80.0 | 63.3 | 72.9 | 74.9 | 61.7 | 49.5 | 79.7 | 70.5 | 53.6 | 82.2 | 67.9 |
| MDD (Zhang et al., 2019) | 54.9 | 73.7 | 77.8 | 60.0 | 71.4 | 71.8 | 61.2 | 53.6 | 78.1 | 72.5 | 60.2 | 82.3 | 68.1 |
| MCC (Jin et al., 2020) | 55.1 | 75.2 | 79.5 | 63.3 | 73.2 | 75.8 | 66.1 | 52.1 | 76.9 | 73.8 | 58.4 | 83.6 | 69.4 |
| DCAN (Li et al., 2020) | 54.5 | 75.7 | 81.2 | 67.4 | 74.0 | 76.3 | 67.4 | 52.7 | 80.6 | 74.1 | 59.1 | 83.5 | 70.5 |
| DALN (Chen et al., 2022) | 57.8 | 79.9 | 82.0 | 66.3 | 76.2 | 77.2 | 66.7 | 55.5 | 81.3 | 73.5 | 60.4 | 85.3 | 71.8 |
| FixBi (Na et al., 2021) | 58.1 | 77.3 | 80.4 | 67.7 | 79.5 | 78.1 | 65.8 | 57.9 | 81.7 | 76.4 | 62.9 | 86.7 | 72.7 |
| DCAN+SCDA (Li et al., 2021b) | 60.7 | 76.4 | 82.8 | 69.8 | 77.5 | 78.4 | 68.9 | 59.0 | 82.7 | 74.9 | 61.8 | 84.5 | 73.1 |
| ATDOC (Liang et al., 2021) | 60.2 | 77.8 | 82.2 | 68.5 | 78.6 | 77.9 | 68.4 | 58.4 | 83.1 | 74.8 | 61.5 | 87.2 | 73.2 |
| EIDCo (Zhang et al., 2023) | 63.8 | 80.8 | 82.6 | 71.5 | 80.1 | 80.9 | 72.1 | 61.3 | 84.5 | 78.6 | 65.8 | 87.1 | 75.8 |
| ECB (Ngo et al., 2024) | 68.5 | 85.4 | 88.3 | 79.2 | 86.8 | **89.0** | 79.3 | 66.4 | 88.5 | 81.0 | 71.1 | 90.4 | 81.2 |
| **ECB-G** | **73.5** | **85.6** | **90.1** | **82.5** | **86.9** | 88.8 | **81.5** | **69.0** | **89.0** | **83.6** | **73.1** | **91.4** | **82.9** |

