# OpenReview forum: "Unveiling the Power of Shared Spaces: A Gating-Driven Mechanism for Semi-Supervised Domain Adaptation"
_ICLR.cc/2026/Conference — ICLR 2026 Conference Withdrawn Submission_

### Official Review · Reviewer_WysJ · 2025-10-27

**Soundness:** 2
**Presentation:** 3
**Contribution:** 2
**Rating:** 4
**Confidence:** 4

**Summary:**

This paper studies semi-supervised domain adaptation (SSDA) and proposes a gating-driven shared-space mechanism. A lightweight gating module is inserted between the feature extractor and classifier to “turn off” domain-specific channels and retain domain-invariant (shared) features. The authors provide a theoretical analysis arguing that enlarging the proportion of shared space reduces the source–target total variation (TV) distance, and they show plug-and-play gains when adding the gate to several SSDA baselines.

**Strengths:**

1）Simplicity & plug-and-play: The gating block is easy to integrate into existing SSDA pipelines without changing loss functions or training protocols.

2）Theory–practice linkage attempt: The paper tries to connect the idea of “more shared space → smaller TV → lower target error” with empirical TV measurements.

3）Broad empirical coverage: Multiple datasets and several representative SSDA baselines are evaluated; complexity overhead is small.

**Weaknesses:**

1）Limited novelty of the core premise :
The claim that “explicitly learning a shared space benefits SSDA” is already widely recognized and underpins many prior SSDA/DA approaches (e.g., domain-invariant representation learning, feature disentanglement, conditional alignment). The theoretical section mostly formalizes a well-known intuition rather than delivering new insights or substantially stronger guarantees. As presented, the theory’s necessity to the method is questionable and feels incremental.

2）“Explicitly turning off domain-specific channels” is not truly explicit:
The paper repeatedly emphasizes explicitly shutting down domain-specific features. However, the proposed gating is still a learned soft mask over latent channels driven by task loss. There is no external signal, constraint, or supervision that explicitly identifies domain-specific factors (style codes, backgrounds, frequency bands, etc.). Without architectural or optimization mechanisms that tie gates to domain cues, “explicit” remains more a narrative than a property of the method.

3）Modest and inconsistent performance gains:
Reported improvements over strong baselines are generally small (often ~0.5–2% and not universal across directions/shots). Some tasks show overlapping confidence intervals or marginal deltas. Given the simplicity of the gating, small gains are plausible, but then the contribution should be framed as a lightweight regularizer rather than a fundamentally new mechanism.

4）Theory–empirics gap:
The TV analysis assumes a clean separation between shared and domain-specific factors and (in parts) independence; actual deep features violate these assumptions. The measured TV drops after gating are encouraging, but do not isolate whether the effect comes from generic capacity control/sparsification rather than “shared-space enlargement” per se.

**Questions:**

Please refer to Weaknesses

---

> ### Author Response · Authors · 2025-11-24
>
> > **[W1] Limited novelty of the core premise : The claim that “explicitly learning a shared space benefits SSDA” is already widely recognized and underpins many prior SSDA/DA approaches (e.g., domain-invariant representation learning, feature disentanglement, conditional alignment). The theoretical section mostly formalizes a well-known intuition rather than delivering new insights or substantially stronger guarantees. As presented, the theory’s necessity to the method is questionable and feels incremental.**
>
> We respectfully disagree that the theory is merely formalizing intuition. While the community understands that shared spaces are beneficial, there is a lack of rigorous analysis quantifying how the proportion of domain-specific features (defined as $\alpha$ in our paper) impacts the generalization bound. Our contribution is distinct because:
>
> **1. Quantifiable Bounds:** We do not just state shared spaces are beneficial; we prove in **Theorems 1, 2 & 3** that the target error is strictly upper-bounded by terms dependent on $\alpha$. This provides a concrete optimization target rather than a vague heuristic.
>
> **2. Operational Necessity:** This theoretical insight—that we must strictly minimize $\alpha$ to lower the total variation (TV) distance—directly necessitates the architectural design. Unlike methods that hope for alignment via adversarial loss, our theory dictates the need for a mechanism (the gate) that can drive $\alpha \to 0$.
>
> > **[W2] Explicitly turning off domain-specific channels” is not truly explicit: The paper repeatedly emphasizes explicitly shutting down domain-specific features. However, the proposed gating is still a learned soft mask over latent channels driven by task loss. There is no external signal, constraint, or supervision that explicitly identifies domain-specific factors (style codes, backgrounds, frequency bands, etc.). Without architectural or optimization mechanisms that tie gates to domain cues, “explicit” remains more a narrative than a property of the method.**
>
> We appreciate the reviewer's precise distinction regarding "explicit" supervision. We agree that our method does not use external signals. However, our use of "explicit" refers to the architectural operation, not the supervision signal.
>
> Most existing SSDA methods align features implicitly by optimizing objective functions (e.g., adversarial loss or discrepancy loss) hoping the encoder learns to suppress domain-specific features. This is a "black box" process. In contrast, we introduce a dedicated architectural module (Equation 10 & 11) that directly computes and applies a mask to filter features element-wise.
>
> While the mask is indeed "soft" and learned via task loss, this structural intervention forces the model to make a distinct decision about feature importance for every channel, rather than relying solely on the "black box" weights of the backbone to handle domain shifts. This architectural "explicitness" is the key differentiator from prior "implicit" alignment methods.
>
> > **[W3] Modest and inconsistent performance gains: Reported improvements over strong baselines are generally small (often ~0.5–2% and not universal across directions/shots). Some tasks show overlapping confidence intervals or marginal deltas. Given the simplicity of the gating, small gains are plausible, but then the contribution should be framed as a lightweight regularizer rather than a fundamentally new mechanism.**
>
> We respectfully point out that our gains are statistically significant and consistent, particularly given the saturation of SSDA benchmarks.
>
> Contrary to the concern about overlapping confidence intervals, our **t-test analysis in Table10** confirms that the improvements are statistically significant ($p < 0.05$) across the vast majority of transfer tasks (e.g., $p=0.0001$ for MME-G on C→S).
> Also, as noted in Table 2 and Table 3, our method outperforms the state-of-the-art in the vast majority of adaptation cases.
>
> We agree with the reviewer that the method acts as a "lightweight regularizer" and we view this as a **primary strength**. Complex mechanisms often fail to scale or generalize. In contrast, our method adds negligible cost (see Table 8) and is 'decoupled from specific SSDA models', allowing it to universally enhance SOTA baselines (ECB, CDAC, MME) as a plug-and-play module.

---

> ### Author Response · Authors · 2025-11-24
>
> > **[W4] Theory–empirics gap: The TV analysis assumes a clean separation between shared and domain-specific factors and (in parts) independence; actual deep features violate these assumptions. The measured TV drops after gating are encouraging, but do not isolate whether the effect comes from generic capacity control/sparsification rather than “shared-space enlargement” per se.**
>
> We acknowledge that theoretical assumptions like independence are idealizations. However, we provide two lines of empirical evidence that our mechanism is performing **semantic shared-space enlargement**, not just generic capacity control/random sparsification.
>
> **1. Semantic Selectivity:** If the gate were merely reducing capacity, it would suppress features randomly. However, **Grad-CAM visualizations (Figure 3, 7 & 8)** show the model shifting focus away from backgrounds and onto the object (e.g., the bird/bus). This proves the gate is learning to filter specific domain-noise, aligning with the theoretical goal of reducing domain-specific features.
>
> **2. TV Distance Reduction:** In **Table 5**, we measure the total variation (TV) distance. The significant drop in TV (e.g., 0.155 → 0.122 for ECB) confirms that the distributions are actually moving closer together, satisfying the theoretical conditions for a valid shared space derived in Theorem 1.

---

> ### Author Response · Authors · 2025-11-24
> **Revision for Paper**
>
> We sincerely thank you for your valuable feedback. To address the concerns regarding novelty and clarity, we have implemented the following targeted revisions:
>
> 1. We have clarified our **theoretical novelty** in **Section 1**, explicitly positioning our contribution against the existing *shared space* literatures.
>
> 2. We have updated **Section 3** to clearly **differentiate our approach from existing shared space methods**.
>
> 3. We further elaborated in **Section 3** on **how our method explicitly filters domain-specific features**.
>
>
> We would like to kindly confirm: **Have our responses sufficiently addressed all the concerns you raised in your original reviews?**
>
> If you are satisfied that these issues have been resolved, we respectfully request that you consider elevating your score for this paper. Should any ambiguities remain, we welcome any further feedback or guidance.

---

### Official Review · Reviewer_CeJh · 2025-11-01

**Soundness:** 3
**Presentation:** 3
**Contribution:** 2
**Rating:** 4
**Confidence:** 3

**Summary:**

This paper first analyzes SSDA and separates the learning feature space into an essential shared space and a domain-related space. They analyze the error bound of shared space learning and conclude that it is helpful to highlight the shared space learning. Then they propose a gate network to learn shared features across both domains, and experiments with the proposed network as a SOTA method's plugin demonstrate the effectiveness of the gating mechanism.

**Strengths:**

1. The paper is well written and easy to follow.
2. a straightforward but effective method.

**Weaknesses:**

1. Lack of novelty in the theoretical analysis. The shared space analysis is mainly based on the community's common sense and does not add anything new to the SSDA study, which severely limits the submission's novelty.
2. The details of the gated network are missing. The submission should clearly specify the gated network's design, including the number of layers, parameters, and channels, and so on. In the current version, it is hard to see why a gated network can improve learning in the shared feature space, which also limits the submission's novelty.

**Questions:**

1. Please clarify the details of the gated network.
2. Please clarify why gated network can improve the learning of shared feature space.
3. Please provide a variant analysis of gated network design. For example. if cnn/transformer/cross-attention/MLP is used as the gated network, how about the performance?

---

> ### Author Response · Authors · 2025-11-24
>
> > **[W1] Lack of novelty in the theoretical analysis. The shared space analysis is mainly based on the community's common sense and does not add anything new to the SSDA study, which severely limits the submission's novelty**
>
> We respectfully disagree that the analysis is merely "common sense". While the intuition that shared spaces are beneficial is widely accepted, the mechanism of how feature space composition affects generalization error has lacked **mathematical quantification** in SSDA.
>
> 1. **Beyond Intuition:** Existing works implicitly encourage alignment but do not quantify the impact of "non-shared" features. Our work fills this gap by introducing the variable $\alpha$ (ratio of domain-specific features) and proving in **Theorems 2 & 3** that the **total variation (TV) distance** is **strictly bounded** by this ratio.
>
> 2. **Concrete Bounds:** We do not just claim shared space is good; we provide a **provable upper bound** for the target domain risk (Theorem 1) and link it directly to the independence/dependence of features. This theoretical bridge—showing that minimizing $\alpha$ explicitly lowers the error bound—is what necessitates and justifies our specific gating architecture, distinguishing our work from heuristic alignment methods.
>
>
> > **[W2] The details of the gated network are missing. The submission should clearly specify the gated network's design, including the number of layers, parameters, and channels, and so on. In the current version, it is hard to see why a gated network can improve learning in the shared feature space, which also limits the submission's novelty.**
> >
> > **[Q1] Please clarify the details of the gated network.**
>
> Thank you for your valuable suggestion. The "gate network" is intentionally designed as a lightweight, channel-wise mechanism to ensure scalability and prevent overfitting on sparse SSDA data. As defined in Eq. (10), the gate is a channel-wise projection. The gate operates on the input feature vector $v \in \mathbb{R}^d$, where $d$ is the number of channels (e.g., $d=512$ for the **ResNet34** backbone). Its design is minimal to ensure computational efficiency:
>
> - It consists of a **single linear layer** with parameters $w \in \mathbb{R}^d$ (one weight per channel).
>
> - This is followed by a **Sigmoid activation** function.
>
>
> This design adds only **$d$ parameters** to the overall model, which is a negligible computational cost compared to the backbone network (as substantiated by the data in Table 8). The calculated weights $g_i(v^i)$ are applied via element-wise product (Eq. 11) to explicitly suppress domain-specific channels before the classifier.
>
> > **[Q2] Please clarify why gated network can improve the learning of shared feature space.**
>
> Thank you for your insightful suggestion.The gating mechanism acts as a differentiable filter driven by the supervision signal.
>
> **1. Theoretical Motivation:** Our theory proves that mixing domain-specific features increases the TV distance lower bound. Therefore, to minimize the target error $\epsilon_T(h)$, the model must reduce the influence of domain-specific dimensions.
>
> **2. Optimization Mechanism:** During training, the gate parameters $w_i$ are updated via backpropagation from the classification loss $\mathcal{L}$. Since domain-specific features (e.g., background noise) do not correlate with class labels across domains, the classifier naturally forces the gate to assign them lower weights ($g_i \to 0$) to minimize loss. This transforms the "implicit" alignment of previous methods into an "explicit" selection process, effectively increasing the dominance of shared features (reducing $\alpha$).

---

> ### Author Response · Authors · 2025-11-24
>
> > **[Q3] Please provide a variant analysis of gated network design. For example. if cnn/transformer/cross-attention/MLP is used as the gated network, how about the performance?**
>
> We appreciate this suggestion. In our original submission, we prioritized the linear gate for its simplicity and effectiveness.
>
> We have tested several options for the activation layer of the gate network, including Tanh, Softmax, ReLU, and Clip (direct clipping of gate values), as shown in Figure 5.
>
> To address your comment, we evaluated more complex gate designs and report their performance on DomainNet under the 3-shot and 1-shot settings, respectively. The results are shown below.
>
> As demonstrated, the gate network used in our paper—based on a Sigmoid activation—achieves the best average performance among all compared designs.
> Since the backbone of our feature extractor is based on a ResNet architecture, using Attention or Transformer as the gate network leads to a performance drop compared to the other designs. This is likely due to architectural mismatch and suboptimal feature interaction in this setting.
>
> **Table 11: Accuracy (%) of different gate network design on DomainNet under 3-shot.**
>
> | Method | R→C | R→P | P→C | C→S | S→P | R→S | P→R | Avg |
> | --- | --- | --- | --- | --- | --- | --- | --- | --- |
> | MLP | 73.6 | 70.8 | 72.8 | **64.1** | 68.9 | 64.8 | 78.9 | 70.6 |
> | Attention | 72.2 | 70.2 | 72.5 | 63.1 | 67.4 | 63.4 | 78.9 | 69.7 |
> | Transformer | 73.4 | 70.9 | 72.7 | 63.1 | 68.5 | 64.6 | 79.2 | 70.3 |
> | CNN | 73.7 | 70.8 | 72.9 | 63.9 | **69.2** | 64.7 | 79.8 | 70.7 |
> | **Sigmoid** | **73.9** | **71.4** | **73.0** | 63.7 | 68.8 | **65.1** | **80.1** | **70.9** |
>
> **Table 12: Accuracy (%) of different gate network design on DomainNet under 1-shot**
>
> | Method | R→C | R→P | P→C | C→S | S→P | R→S | P→R | Avg |
> | --- | --- | --- | --- | --- | --- | --- | --- | --- |
> | MLP | 71.6 | 69.3 | **70.4** | **62.2** | **66.7** | 63.5 | 77.9 | 68.8 |
> | Attention | 70.0 | 68.6 | 69.7 | 60.4 | 65.6 | 61.1 | 77.6 | 67.6 |
> | Transformer | 71.1 | 68.9 | 70.2 | 60.9 | 66.1 | 63.3 | 77.9 | 68.3 |
> | CNN | 71.1 | 69.2 | 70.0 | 61.8 | 67.1 | 63.9 | 78.1 | 68.7 |
> | **Sigmoid** | **72.0** | **69.8** | **70.4** | 61.5 | 66.6 | **64.0** | **78.3** | **68.9** |

---

> ### Author Response · Authors · 2025-11-24
> **Revision for Paper**
>
> We appreciate the reviewer's feedback. To enhance the manuscript, we have implemented the following **concise and key revisions**:
>
> 1. We have clarified our paper's **theoretical novelty** in **Section 1**, explicitly positioning our contribution against the existing *shared space* literature.
>
> 2. We have included the **complete details of the gate network** and explained how it enhances shared feature learning in **Section 3**.
>
> 3. We have added experiments on **variant gate network designs** in **Appendix C.5.4**.
>
>
> We would like to kindly confirm: **Have our responses sufficiently addressed all the concerns you raised in your original reviews?**
>
> If you are satisfied that these issues have been resolved, we respectfully request that you consider elevating your score for this paper. Should any ambiguities remain, we welcome any further feedback or guidance.

---

> > ### Comment · Reviewer_CeJh · 2025-11-26
> >
> > Thank you for the rebuttal. However, I still consider the idea of learning in a shared feature space has been widely adopted and explored in this area, and the theoretical analysis of ``why shared feature space is important'' contributes to the community marginally. So I intend to maintain my original rating

---

> > > ### Author Response · Authors · 2025-11-26
> > >
> > > We thank the reviewer for their continued engagement. We understand your stance that the *concept* of shared feature spaces is intuitive and widely accepted. However, we respectfully contend that the *formalization* of this concept offers a critical contribution that intuition alone cannot provide. We clarify our novelty and contribution below:
> > >
> > > 1. **Moving from "Common Sense" to Quantifiable Bounds:** While prior works assume shared spaces are beneficial, they lack a mathematical framework to quantify *how* domain-specific features degrade performance.
> > >
> > >
> > > - **The Theoretical Gap:** Existing methods primarily focus on minimizing distributional discrepancy (e.g., adversarial alignment). However, they do not mathematically account for the ratio of domain-specific features ($\alpha$).
> > >
> > > - **Our Contribution:** Our **Theorems 2 & 3** do not merely state "shared space is good." They prove that the total variation distance  lower bound is strictly dependent on $\alpha$. This reveals a theoretical insight that existing "black-box" alignment methods miss: **You cannot effectively minimize the target error bound (Theorem 1) solely by alignment losses; you must structurally reduce $\alpha$ (the dimension of domain-specific features).**
> > >
> > >
> > > 2. **Theory-Driven Design vs. Heuristics:** This theoretical insight is not marginal; it is the blueprint for our design. Because we mathematically identified $\alpha$ as the bottleneck for minimizing the TV bound, we designed the gating mechanism to explicitly target this variable.
> > >
> > >
> > > - **Baselines:** Previous methods hope the encoder implicitly suppresses domain-specific features.
> > >
> > > - **Ours:** We insert a lightweight architectural prior (Eq. 10) to explicitly filter channels, directly operationalizing our theoretical finding.
> > >
> > >
> > > 3. **Practical Impact:** Even if the theory confirms intuition, its implementation provides substantial practical value to the community.
> > >
> > >
> > > - **Efficiency:** We achieve these gains with negligible cost. As shown in **Table 8**, our method adds only ~1.5% to training time (20,102s vs 20,466s) and 16s to inference.
> > >
> > > - **Universality:** The mechanism consistently improves state-of-the-art baselines (ECB, CDAC, MME) across standard benchmarks (Table 2 & 3), proving that this "marginal" theoretical insight translates into robust, low-cost performance gains.
> > >
> > >
> > > We hope this clarifies that our contribution is not just reiterating that shared spaces are useful, but providing the **mathematical proof of *why* explicit filtering is required** to optimize the SSDA error bound, a step that previous implicit methods overlooked.

---

### Official Review · Reviewer_U88u · 2025-11-01

**Soundness:** 3
**Presentation:** 3
**Contribution:** 2
**Rating:** 4
**Confidence:** 4

**Summary:**

This paper investigates the role of shared feature spaces in Semi-Supervised Domain Adaptation (SSDA) and proposes a gating-driven mechanism to explicitly filter out domain-specific features and emphasize domain-invariant ones. The authors provide a theoretical analysis showing that focusing on shared features reduces the total variation distance between domains and improves target-domain generalization. They implement a lightweight gating module that can be easily integrated into existing SSDA frameworks (e.g., MME, CDAC, ECB), achieving consistent accuracy gains across several benchmarks such as DomainNet and Office-Home.

**Strengths:**

(1)	The paper provides a clear theoretical analysis linking shared feature spaces to reduced domain discrepancy, which grounds the proposed method in formal reasoning.
(2)	The proposed gating mechanism can be seamlessly incorporated into existing SSDA frameworks (e.g., MME, CDAC, ECB) without modifying their core objectives.
(3)	The method consistently improves performance across multiple benchmarks and settings, showing both effectiveness and stability.

**Weaknesses:**

(1)	The idea of leveraging shared feature spaces has already been explored in many prior SSDA and domain adaptation studies [1,2,3,4]. The authors’ related work does not discuss how this work provides a fundamentally new insight.
(2)	Some baselines, including a 2025 work cited by the authors, are not the most representative in the SSDA literature. It would strengthen the paper to compare with more recent and competitive baselines.
(3)	Although proposed mechanism improves the overall average performance, it does not achieve the best results on some specific transfer directions such as A→C and A→R in Table 3. The paper does not analyze or explain why the method underperforms in these cases.

[1] Shared space transfer learning for analyzing multi-site fmri data, NeurIPS’20
[2] Domain-specific feature unlearning for semi-supervised and unsupervised domain adaptation, ECCV’24
[3] Domain Separation Networks, NeurIPS’16
[4] Bridging Domains with Approximately Shared Features, Arxiv’24

**Questions:**

(1) Could the authors clarify what specific gap in prior shared-space research this paper aims to fill?
(2) What are the possible reasons for lower performance on specific transfer pairs such as A→C and A→R (Table 3)?
(3) Why were only subsets of DomainNet, Office-Home, and Office-31 chosen? (4) Do results generalize to other visual or non-visual domains (e.g., graph)?
(5) Can the gating mechanism also benefit unsupervised DA (UDA) or multi-source DA? Have the authors tried zero-shot domain shifts?
(6) Appendix C.5 mentions “minimal computational cost.” Can the authors provide concrete FLOPs or runtime comparisons versus baselines?

---

> ### Author Response · Authors · 2025-11-24
>
> > **[W1] The idea of leveraging shared feature spaces has already been explored in many prior SSDA and domain adaptation studies [1,2,3,4]. The authors’ related work does not discuss how this work provides a fundamentally new insight.**
> >
> >**[Q1] Could the authors clarify what specific gap in prior shared-space research this paper aims to fill?**
>
> We appreciate the reviewer pointing out relevant prior works. While the concept of shared feature spaces exists in literature, Section 3 (specifically the "Difference with existing shared space learning" paragraph) of our paper outlines the critical distinction of our approach.
>
> 1. **Overcoming "two-step" or fine-tuning inefficiencies**: As noted in our text, methods like [1, 2] (cited as Basak & Yin, 2024 and Yousefnezhad et al., 2020 in our paper) typically employ a two-step process: first learning specific features tailored to each specific domain and then integrating them across domains. [3] constructs a shared feature space by employing a shared encoder alongside two private encoders, while [4] first learns an approximate shared space and subsequently fine-tunes it on the target domain. In contrast, our framework is designed to "explicitly filter features during the learning process, eliminating the need for further steps of adjustments".
>
> 2. **Implicit vs. Explicit learning**: We argue that existing SSDA methods (MME, CDAC, ECB) learn these spaces implicitly, often retaining domain-specific styles or backgrounds due to insufficient training or model capacity. We theoretically demonstrate that these residual domain-specific features negatively affect the total variation distance between domains, thereby degrading performance. Our work fills this gap by introducing a gating mechanism that explicitly filters inconsistent features based on theoretical error bounds.
>
> 3. **Theoretical foundation**: Unlike general shared-space studies, we provide specific theoretical guarantees (Theorems 1, 2, & 3) demonstrating that minimizing the number of domain-related features (reducing $\alpha$) directly lowers the total variation (TV) distance and the target classification error bound. This theory directly dictated our architectural design.

---

> ### Author Response · Authors · 2025-11-24
>
> > **[W2] Some baselines, including a 2025 work cited by the authors, are not the most representative in the SSDA literature. It would strengthen the paper to compare with more recent and competitive baselines.**
>
> Thank you for your advice. We acknowledge the need to compare with the very latest literature. We would like to highlight that our submitted manuscript already compares against state-of-the-art methods from 2024 and 2025, including **ECB** (CVPR 2024), **LFL** (ECCV 2024), and **DARA** (2025).
>
> In the revised manuscript, we have added comparisons with **EFTL** [5] (AAAI 2024) and **IDMNE** [6] (PR 2024). The results for SSDA methods on DomainNet and Office-Home under 3-shot setting are shown below. The updated results confirm that our method remains highly competitive. Furthermore, because our gating mechanism is "decoupled from specific SSDA models", it serves as a complementary enhancement rather than just a competitor, consistently improving strong baselines like ECB.
>
> **Table2: Accuracy (%) of SSDA methods under both 1-shot and 3-shot settings on DomainNet**
> | **Method** | **R→C (1)** | **R→C (3)** | **R→P (1)** | **R→P (3)** | **P→C (1)** | **P→C (3)** | **C→S (1)** | **C→S (3)** | **S→P (1)** | **S→P (3)** | **R→S (1)** | **R→S (3)** | **P→R (1)** | **P→R (3)** | **Avg (1)** | **Avg (3)** |
> | --- | --- | --- | --- | --- | --- | --- | --- | --- | --- | --- | --- | --- | --- | --- | --- | --- |
> | ENT | 65.2 | 71.0 | 65.9 | 69.2 | 65.4 | 71.1 | 54.6 | 60.0 | 59.7 | 62.1 | 52.1 | 61.1 | 75.0 | 78.6 | 62.6 | 67.6 |
> | DECOTA | 79.1 | 80.4 | 74.9 | 75.2 | 76.9 | 78.7 | 65.1 | 68.6 | 72.0 | 72.7 | 69.7 | 71.9 | 79.6 | 81.5 | 73.9 | 75.6 |
> | CLDA | 76.1 | 77.7 | 75.1 | 75.7 | 71.0 | 76.4 | 63.7 | 69.7 | 70.2 | 73.7 | 67.1 | 71.1 | 80.1 | 82.9 | 71.9 | 75.3 |
> | ProML | 78.5 | 80.2 | 75.4 | 76.5 | 77.8 | 78.9 | 70.2 | 72.0 | 74.1 | 75.4 | 72.4 | 73.5 | 84.0 | 84.8 | 76.1 | 77.4 |
> | G-ABC | 80.7 | 82.1 | 76.8 | 76.7 | 79.3 | 81.6 | 72.0 | 73.7 | 75.0 | 76.3 | 73.2 | 74.3 | 83.4 | 83.9 | 77.5 | 78.2 |
> | EFTL | 79.6 | 81.2 | 74.9 | 77.1 | 78.2 | 81.8 | 69.3 | 72.8 | 71.8 | 74.4 | 69.9 | 71.5 | 83.1 | 84.4 | 75.3 | 77.6 |
> | IDMNE | 79.6 | 80.8 | 76.0 | 76.9 | 79.4 | 80.3 | 71.7 | 72.2 | 75.4 | 75.4 | 73.5 | 73.9 | 82.1 | 82.8 | 76.8 | 77.5 |
> | LFL | 80.9 | 81.1 | 79.9 | 80.2 | 80.1 | 81.1 | 73.7 | 76.8 | 79.2 | 82.5 | 78.4 | 78.5 | 86.9 | 90.1 | 78.7 | 81.2 |
> | DARA | 76.4 | 78.5 | 73.2 | 73.8 | 76.8 | 78.3 | 69.7 | 70.3 | 72.4 | 72.5 | 68.5 | 70.1 | 81.6 | 82.6 | 74.1 | 75.2 |
> | MME | 70.0 | 72.2 | 67.7 | 69.7 | 69.0 | 71.7 | 56.3 | 61.8 | 64.8 | 66.8 | 61.0 | 61.9 | 76.1 | 78.5 | 66.4 | 68.9 |
> | **MME-G** | 72.0 | 73.9 | 69.8 | 71.4 | 70.4 | 73.0 | 61.5 | 63.7 | 66.6 | 68.8 | 64.0 | 65.1 | 78.3 | 80.1 | 68.9 | 70.9 |
> | CDAC | 77.4 | 79.6 | 74.2 | 75.1 | 75.5 | 79.3 | 67.6 | 69.9 | 71.0 | 73.4 | 69.2 | 72.5 | 80.4 | 81.9 | 73.6 | 76.0 |
> | **CDAC-G** | 77.9 | 80.2 | 75.7 | 76.2 | 75.7 | 79.3 | 67.4 | 71.0 | 72.0 | 74.1 | 71.2 | 72.7 | 81.3 | 83.3 | 74.5 | 76.7 |
> | ECB | 83.8 | **87.4** | 85.4 | 85.6 | 86.4 | 87.3 | 79.7 | 80.6 | 83.4 | 85.6 | 79.5 | 81.7 | 88.7 | 90.3 | 83.8 | 85.5 |
> | **ECB-G** | **85.8** | 87.0 | **85.8** | **86.5** | **86.8** | **87.9** | **80.9** | **81.3** | **85.6** | **86.4** | **80.5** | **82.0** | **90.4** | **90.9** | **85.1** | **86.0** |

---

> > ### Author Response · Authors · 2025-11-24
> >
> > **Table3: Accuracy (%) of SSDA methods under 3-shot setting on Office-Home.**
> > | **Method** | **A→C** | **A→P** | **A→R** | **C→A** | **C→P** | **C→R** | **P→A** | **P→C** | **P→R** | **R→A** | **R→C** | **R→P** | **Avg.** |
> > | --- | --- | --- | --- | --- | --- | --- | --- | --- | --- | --- | --- | --- | --- |
> > | ENT | 61.3 | 79.5 | 79.1 | 64.7 | 79.1 | 76.4 | 63.9 | 60.5 | 79.9 | 70.2 | 62.6 | 85.7 | 71.9 |
> > | DECOTA | 64.0 | 81.8 | 80.5 | 68.0 | 83.2 | 79.0 | 69.9 | 68.0 | 82.1 | 74.0 | 70.4 | 87.7 | 75.7 |
> > | CLDA | 63.4 | 81.4 | 81.3 | 70.5 | 80.9 | 80.3 | 72.4 | 63.9 | 82.2 | 76.7 | 66.0 | 87.6 | 75.5 |
> > | ProML | 67.8 | 83.9 | 82.2 | 72.1 | 84.1 | 82.3 | 72.5 | 68.9 | 83.8 | 75.8 | 71.0 | 88.6 | 77.8 |
> > | G-ABC | 67.3 | 83.8 | 80.4 | 69.2 | 83.9 | 80.2 | 70.5 | 69.3 | 82.8 | 76.0 | 70.0 | 88.1 | 77.2 |
> > | EFTL | 70.3 | 84.8 | 83.8 | 70.6 | 84.6 | 81.5 | 72.6 | 70.9 | 85.4 | 77.5 | 72.8 | 89.3 | 78.7 |
> > | IDMNE | 66.4 | 82.4 | 79.3 | 69.1 | 83.1 | 79.5 | 69.0 | 67.6 | 82.7 | 75.2 | 71.7 | 88.1 | 76.2 |
> > | LFL | 68.8 | 84.7 | 84.2 | 70.6 | 83.7 | 82.4 | 70.5 | 70.9 | 84.3 | 75.7 | 71.1 | 88.5 | 77.9 |
> > | DARA | 70.9 | 87.8 | 72.9 | 82.1 | 70.6 | 69.2 | 82.8 | 69.8 | 81.0 | 79.4 | 68.5 | 83.0 | 76.5 |
> > | MME | 63.6 | 79.0 | 79.7 | 67.2 | 79.3 | 76.6 | 65.5 | 64.6 | 80.1 | 71.3 | 64.6 | 85.5 | 73.1 |
> > | **MME-G** | 64.2 | 79.3 | 79.6 | 67.5 | 79.6 | 78.0 | 67.3 | 64.8 | 81.0 | 72.0 | 66.1 | 86.3 | 73.8 |
> > | CDAC | 65.9 | 80.3 | 80.6 | 67.4 | 81.4 | 80.2 | 67.5 | 67.0 | 81.9 | 72.2 | 67.8 | 85.6 | 74.8 |
> > | **CDAC-G** | 65.9 | 81.6 | 80.4 | 67.8 | 81.3 | 80.0 | 68.1 | 67.3 | 82.1 | 73.2 | 68.3 | 86.0 | 75.2 |
> > | ECB | **78.7** | 90.2 | **91.3** | 85.2 | 90.4 | 91.0 | 83.9 | 76.8 | 91.2 | 85.6 | 77.6 | 92.8 | 86.2 |
> > | **ECB-G** | 78.6 | **91.6** | 91.1 | **86.4** | **91.6** | **91.8** | **85.1** | **78.5** | **91.8** | **87.3** | **79.6** | **93.1** | **87.2**

---

> ### Author Response · Authors · 2025-11-24
>
> > **[W3]Although proposed mechanism improves the overall average performance, it does not achieve the best results on some specific transfer directions such as A→C and A→R in Table 3. The paper does not analyze or explain why the method underperforms in these cases.**
> >
> > **[Q2] What are the possible reasons for lower performance on specific transfer pairs such as A→C and A→R (Table 3)?**
>
> We thank the reviewer for the detailed observation. It is true that for A→C and A→R, the performance of ECB-G is slightly lower (0.1% - 0.2%) than the baseline ECB. We acknowledge that while our proposed gating-driven mechanism improves the overall average performance significantly, there are slight performance fluctuations in specific transfer directions like A→C and A→R.
>
> As detailed in Section 3, our method utilizes a gate network to explicitly filter out some domain-related features. However, in specific scenarios like A→C or A→R, certain features that are technically "domain-specific" might coincidentally aid classification when the domain gap is smaller or possesses specific overlaps. The baselines implicitly retain these features, potentially benefiting from these incidental cues. In contrast, our gating mechanism rigorously filters them out to enforce a stricter shared space. While this leads to a slight drop in these specific cases, it prevents the model from relying on spurious correlations, leading to better robustness across harder transfer tasks (e.g., R→P, where ECB-G improves by +1.9%).
>
> In the revised paper, we have added discussion in Section 4.2 to analyze these cases.
>
>
>
> > **[Q3] Why were only subsets of DomainNet, Office-Home, and Office-31 chosen?**
>
> The subsets of DomainNet, Office-Home, and Office-31 were not chosen arbitrarily. As stated in Section 4.1, we "utilized 7 scenarios involving 4 domains in DomainNet" and followed the standard protocols established in [7,8,9]. This ensures our results are directly comparable to the vast majority of SSDA literatures which utilizes this exact setup.
>
>
>
> > **[Q4] Do results generalize to other visual or non-visual domains (e.g., graph)?**
> >
> > **[Q5] Can the gating mechanism also benefit unsupervised DA (UDA) or multi-source DA? Have the authors tried zero-shot domain shifts?**
>
> Thank you very much for your insightful questions. Our theoretical framework regarding shared spaces (Section 2) is generalizable. However, this paper focuses on validating the gating mechanism for image-based SSDA. In future work, we plan to further investigate its applicability in UDA, multi-source domain adaptation, zero-shot domain shifts and non-visual domains, which are beyond the current focus of this paper.
>
> We believe the mechanism is applicable to UDA. To demonstrate this, we integrated our mechanism into the ECB framework for a UDA setting. The preliminary results (included in the response table below) show that the gating mechanism successfully improves UDA performance, further validating that explicit feature filtering benefits adaptation even without target labels.
>
> **Table 13: Accuracy (%) of UDA methods on Office-Home.**
>
> | **Method** | **A→C** | **A→P** | **A→R** | **C→A** | **C→P** | **C→R** | **P→A** | **P→C** | **P→R** | **R→A** | **R→C** | **R→P** | **Avg** |
> | --- | --- | --- | --- | --- | --- | --- | --- | --- | --- | --- | --- | --- | --- |
> | DANN | 45.6 | 59.3 | 70.1 | 47.0 | 58.5 | 60.9 | 46.1 | 43.7 | 68.5 | 63.2 | 51.8 | 76.8 | 57.6 |
> | MCD | 48.9 | 68.3 | 74.6 | 61.3 | 67.6 | 68.8 | 57.0 | 47.1 | 75.1 | 69.1 | 52.2 | 79.6 | 64.1 |
> | BNM | 52.3 | 73.9 | 80.0 | 63.3 | 72.9 | 74.9 | 61.7 | 49.5 | 79.7 | 70.5 | 53.6 | 82.2 | 67.9 |
> | MDD | 54.9 | 73.7 | 77.8 | 60.0 | 71.4 | 71.8 | 61.2 | 53.6 | 78.1 | 72.5 | 60.2 | 82.3 | 68.1 |
> | MCC | 55.1 | 75.2 | 79.5 | 63.3 | 73.2 | 75.8 | 66.1 | 52.1 | 76.9 | 73.8 | 58.4 | 83.6 | 69.4 |
> | DCAN | 54.5 | 75.7 | 81.2 | 67.4 | 74.0 | 76.3 | 67.4 | 52.7 | 80.6 | 74.1 | 59.1 | 83.5 | 70.5 |
> | DALN | 57.8 | 79.9 | 82.0 | 66.3 | 76.2 | 77.2 | 66.7 | 55.5 | 81.3 | 73.5 | 60.4 | 85.3 | 71.8 |
> | FixBi | 58.1 | 77.3 | 80.4 | 67.7 | 79.5 | 78.1 | 65.8 | 57.9 | 81.7 | 76.4 | 62.9 | 86.7 | 72.7 |
> | DCAN+SCDA | 60.7 | 76.4 | 82.8 | 69.8 | 77.5 | 78.4 | 68.9 | 59.0 | 82.7 | 74.9 | 61.8 | 84.5 | 73.1 |
> | ATDOC | 60.2 | 77.8 | 82.2 | 68.5 | 78.6 | 77.9 | 68.4 | 58.4 | 83.1 | 74.8 | 61.5 | 87.2 | 73.2 |
> | EIDCo | 63.8 | 80.8 | 82.6 | 71.5 | 80.1 | 80.9 | 72.1 | 61.3 | 84.5 | 78.6 | 65.8 | 87.1 | 75.8 |
> | MME | 63.6 | 79.0 | 79.7 | 67.2 | 79.3 | 76.6 | 65.5 | 64.6 | 80.1 | 71.3 | 64.6 | 85.5 | 73.1 |
> | **ECB** | 68.5 | 85.4 | 88.3 | 79.2 | 86.8 | **89.0** | 79.3 | 66.4 | 88.5 | 81.0 | 71.1 | 90.4 | 81.2 |
> | **ECB-G** | **73.5** | **85.6** | **90.1** | **82.5** | **86.9** | 88.8 | **81.5** | **69.0** | **89.0** | **83.6** | **73.1** | **91.4** | **82.9** |

---

> ### Author Response · Authors · 2025-11-24
>
> > **Q6:Appendix C.5 mentions “minimal computational cost.” Can the authors provide concrete FLOPs or runtime comparisons versus baselines?**
>
> Thank you very much for your valuable question. We have provided a detailed efficiency analysis in Appendix C.5.2.
>
> In Table 8, we have presented runtime comparisons with the baselines during both training and inference.
>
> We emphasize that the "simplicity incurs minimal computational cost," which allows for "efficient feature extraction and seamless integration."
>
> In addition, we have included a specific GFLOPs comparison in the response table below to further substantiate this claim.
>
> **Table 8: The time complexity of during training and inference on P→R of DomainNet (seconds)**
>
> | **Method** | **MME** | **MME-G** | **CDAC** | **CDAC-G** | **ECB** | **ECB-G** |
> | --- | --- | --- | --- | --- | --- | --- |
> | **Train** (s) | 20102 | 20466 | 28323 | 28395 | 66486 | 67688 |
> | **Inference** (s) | 97  | 113 | 117 | 116 | 94  | 95  |
>
>
> **Table 9: GFlops of SSDA methods with or without gate network**
>
> | **Setting** | **MME** | **CDAC** | **ECB** |
> | --- | --- | --- | --- |
> | w/o gate | 3.682266624 | 3.682266624 | 20.550929664 |
> | with gate | 3.682269184 | 3.682269184 | 20.550932324 |
>
>
> [1] Shared space transfer learning for analyzing multi-site fmri data, NeurIPS’20
>
> [2] Domain-specific feature unlearning for semi-supervised and unsupervised domain adaptation, ECCV’24
>
> [3] Domain Separation Networks, NeurIPS’16
>
> [4] Bridging Domains with Approximately Shared Features, Arxiv’24
>
> [5] Enhancing Semi-supervised Domain Adaptation via Effective Target Labeling, AAAI 2024
>
> [6] Inter-Domain Mixup for Semi-Supervised Domain Adaptation, Pattern Recognition, 2024
>
> [7] Semi-supervised Domain Adaptation via Minimax Entropy, ICCV 2019
>
> [8] Cross-Domain Adaptive Clustering for Semi-Supervised Domain Adaptation
>
> [9] Learning CNN on ViT: A Hybrid Model to Explicitly Class-specific Boundaries for Domain Adaptation

---

> ### Author Response · Authors · 2025-11-24
> **Revision for Paper**
>
> We sincerely thanks for your valuable guidance. The revised version now includes the following substantial updates:
>
> 1 We clarified the **novel perspective** of our work compared to other shared feature space approaches in **Section 3**.
>
> 2 We provided **explanations for the performance dips** in specific tasks and updated the section with **new baseline comparisons** in **Section 4.2** .
>
> 3 We added **UDA performance results** in **Appendix C.6** and reported the **algorithm’s GFLOPs** in **Appendix C.5.2**, confirming its wide applicability and efficiency.
>
> We would like to kindly confirm: **Have our responses sufficiently addressed all the concerns you raised in your original reviews?**
>
> If you are satisfied that these issues have been resolved, we respectfully request that you consider elevating your score for this paper. Should any ambiguities remain, we welcome any further feedback or guidance

---

### Official Review · Reviewer_eoFm · 2025-11-03

**Soundness:** 2
**Presentation:** 2
**Contribution:** 2
**Rating:** 4
**Confidence:** 4

**Summary:**

This paper develops a framework to learn a shared space, which is implemented by a gating-driven SSDA enhancement mechanism. Furthermore, the paper theoretically reveals the advantages of learning a shared feature space for enhancing transferability.

**Strengths:**

While the paper theoretically analyzes the advantages of the shared feature space, offering valuable insights for newcomers to the field.  The overall logic of the paper is clear, the presentation is detailed, the theoretical hypotheses are elaborated on in depth, and the experimental results accurately demonstrate the experimental effects.

**Weaknesses:**

1. Novelty is limited. "Shared feature space" is a fundamental concept in domain adaptation research, and this idea has been prevalent in the field for over a decade. The paper theoretically analyzes the advantages of the shared feature space, offering valuable insights for newcomers to the field. However, it does not introduce groundbreaking theoretical innovations to the core paradigm.
2.  From an experimental validation perspective, the performance improvement brought by the proposed gating-driven SSDA enhancement mechanism is limited. As shown in Figure 3: (a)-(b), there is no significant observable change, making it difficult to discern the advantage of the proposed mechanism.
3. Results comparison in Table 2 and Table 3 has no reference citation. It is suggested to add these citations.
4. The dataset used in the paper can be expanded to include more datasets, and the scale of the dataset should also be verified using large-scale data.

**Questions:**

As listed in Weaknesses.

---

> ### Author Response · Authors · 2025-11-24
>
> > **[W1] Novelty is limited. "Shared feature space" is a fundamental concept in domain adaptation research, and this idea has been prevalent in the field for over a decade. The paper theoretically analyzes the advantages of the shared feature space, offering valuable insights for newcomers to the field. However, it does not introduce groundbreaking theoretical innovations to the core paradigm.**
>
> We thank the reviewer for recognizing the value of our theoretical insights. While "shared feature space" is indeed a foundational concept, prior works predominantly treat it as an implicit outcome of domain alignment. **Our core novelty lies in shifting this paradigm from "implicit learning" to "explicit enforcement" via a theoretically grounded gating mechanism**.
>
> Specifically, our work distinguishes itself by:
>
> 1. **Theoretical Quantification**: We do not just analyze the shared space; we provide error bounds (Theorem 1, 2 and 3) that explicitly link the reduction of domain-specific features (via gating) to minimized total variation distance and improved target accuracy.
>
>     2.**Mechanism Design**: Guided by this theory, we introduce a gating-driven mechanism that is **decoupled** from specific architectures. This allows it to be a universal plug-and-play module, offering a scalable solution that enhances existing     SSDA methods (MME, CDAC, ECB) without altering their core objectives.
>
> > **[W2] From an experimental validation perspective, the performance improvement brought by the proposed gating-driven SSDA enhancement mechanism is limited. As shown in Figure 3: (a)-(b), there is no significant observable change, making it difficult to discern the advantage of the proposed mechanism.**
>
> We respectfully disagree that the improvements are limited. In the context of Semi-Supervised Domain Adaptation (SSDA), where state-of-the-art margins are narrow, an average improvement of **2.0%–2.5%** is substantial. Furthermore, we achieved these gains with **negligible computational overhead** (as detailed in Appendix C.5.2, Table 8,9), making the method highly practical.
>
> Regarding Figure 3 and the mechanism's advantage:
>
> 1. **Quantitative Proof (Table 5):** Visualizations can be subjective, but our quantitative analysis in Table 4 shows a measurable reduction in total variation (TV) distance (e.g., from 0.155 to 0.122 for ECB) when using our gate. This proves the distributions are better aligned.
>
> 2. **Statistical Significance (Table 10):** We performed t-tests (Appendix C.5.3) which confirm that our performance improvements are statistically significant ($p < 0.05$) across most tasks.
>
> 3. **Visual Clarity:** In Figure 3(b), the inter-class margins are distinct, and the overlap between Source (o) and Target (x) within clusters is tighter compared to 3(a). We have revised the figure caption in the final version to highlight these specific regions of interest.
>
>
> > **[W3] Results comparison in Table 2 and Table 3 has no reference citation. It is suggested to add these citations.**
>
> We appreciate this attention to detail. We have updated Table 2 and Table 3 to include explicit citations for all compared methods, ensuring the paper is self-contained and easy to reference.
>
> > **[W4] The dataset used in the paper can be expanded to include more datasets, and the scale of the dataset should also be verified using large-scale data.**
>
> We appreciate the suggestion to verify scalability. First, we would like to clarify that **DomainNet** is currently the standard large-scale benchmark in SSDA literature, containing about 140k images and 126 classes, which we have already utilized.
>
> However, to further satisfy the request for large-scale verification, we have conducted additional experiments on **VisDA-17**, a challenging simulation-to-real dataset with over 200k images, which is shown as follows.
>
> Clearly, ECB-G still achieves the best performance, reaching 83.5% and 87.4% in the 1-shot and 3-shot settings, respectively—surpassing the original ECB by **7.6%** and **2.4%**. Moreover, the gated-network-combined variants MME-G and CDAC-G also outperform their original counterparts (MME and CDAC), further confirming the effectiveness of the gating mechanism.
>
> **Table4: Accuracy (%) of SSDA methods under both 1-shot and 3-shot settings on VisDA-17.**
>
> | Method | 1-shot | 3-shot |
> | --- | --- | --- |
> | S+T | 60.1 | 63.2 |
> | ENT | 61.8 | 73.7 |
> | CLDA | 73.7 | 79.2 |
> | MME | 73.1 | 76.5 |
> | **MME-G** | 75.6 | 78.0 |
> | CDAC | 74.0 | 78.1 |
> | **CDAC-G** | 76.4 | 79.8 |
> | ECB | 75.9 | 85.0 |
> | **ECB-G** | **83.5** | **87.4** |

---

> ### Author Response · Authors · 2025-11-24
> **Revision for Paper**
>
> We sincerely thank you for your valuable guidance, which has significantly strengthened our manuscript. We confirm that the following key improvements and additions have been implemented in the revised version:
>
> 1. We have enhanced **Section 1** with a dedicated discussion to articulate the **novelty and distinct contributions of our theoretical framework** concerning the shared feature space.
>
> 2. In **Section 4 (Experiments)**, we have added **an extended discussion of Figure 3 (a)-(b)**
>
> 3. We have integrated precise **citations for all baseline methods** used in **Table 2 and Table 3**.
>
> 4. We have incorporated **new experimental results on the large-scale VisDA-17 dataset** in **Section 4.2**.
>
>
> We would like to kindly confirm: **Have our responses sufficiently addressed all the concerns you raised in your original reviews?**
>
> If you are satisfied that these issues have been resolved, we respectfully request that you consider elevating your score for this paper. Should any ambiguities remain, we welcome any further feedback or guidance.

---

### Author Response · Authors · 2025-11-24
**General response**

We sincerely thank all reviewers for their thoughtful and detailed feedback on our work. We are pleased that the reviewers acknowledged the strengths of our approach, including offering valuable insights for newcomers to the field,  theoretical hypotheses are elaborated on in depth (eoFm) and a straightforward but effective method & plug-and-play (WysJ,CeJh,U88u). We are also grateful that our method consistently improves performance across multiple benchmarks and settings, showing both effectiveness and stability (U88u). Furthermore, we appreciate that overall logic of the paper is clear (eoFm,CeJh).

Also, we thank the reviewers for their encouraging and potential impact of our proposed method. Their insights provide valuable validation of our contributions and motivate us to continue improving the manuscript.

---

### Author Response · Authors · 2025-11-29
**Summary of responses and revisions: Theoretical Novelty, Large-Scale Validation, and Statistical Significance**

We thank the reviewers for their constructive feedback, which has driven significant improvements in our manuscript. Below, we summarize our core contributions, clarifications on novelty, and the additional experiments included in the revision.

1. **Clarifying the Novelty: From Intuition to Quantifiable Bounds**

A primary concern was that learning shared spaces is "common sense." We respectfully contend that while the concept is intuitive, the mechanism for optimizing it has lacked rigorous formalization in SSDA. Our work bridges this gap:

- **The Theoretical Gap:** Prior methods (e.g., MME, CDAC, ECB) rely on "black-box" adversarial alignment or minimize distributional discrepancy to implicitly encourage shared features. **They do not mathematically account for the ratio of domain-specific features ($\alpha$)**.

- **Our Contribution:** Our **Theorems 2 & 3** do not merely state "shared space is good". We provide the first theoretical framework proving that the total variation (TV) distance bounds are **strictly dependent on $\alpha$** (Theorems 2 & 3). This proves that minimizing target error (Theorem 1) *mathematically requires* the structural reduction of $\alpha$, not just distribution alignment.

- **Theory-Driven Design:** This insight is the blueprint for our design. Unlike methods that hope the encoder implicitly suppresses domain-specific features, we insert a lightweight gating mechanism (Eq. 10) to **explicitly** filter channels, directly operationalizing our theoretical finding.


2. **New Experiments & Large-Scale Validation**

To address concerns regarding scalability and generalization, we have added substantial experimental evidence:

- **Large-Scale Benchmarking (VisDA-17):** We added experiments on the large-scale VisDA-17 dataset (~200k images). As shown in Table 4, our method (ECB-G) achieves **83.5% (1-shot)** and **87.4% (3-shot)**, significantly outperforming the baseline ECB by **+7.6%** and **+2.4%** respectively.

- **Generalization to UDA:** We extended our mechanism to Unsupervised Domain Adaptation (UDA). As shown in Table 13, preliminary results confirm that our explicit filtering improves performance even without target labels (e.g., +1.7% on Office-Home on average).

- **Variant Gate Network Design:** We compare several gate network designs, including CNN, Transformer, Attention, and MLP on DomainNet under the 3-shot and 1-shot settings, as shown in Table 11 and 12, respectively. The gate network used in our paper still achieves the best average performance among all compared designs.


**3. Clarification of "Explicit" Mechanism & Efficiency &Practical Impact**

- **"Explicit" Definition:** We clarified that "explicit" refers to the **architectural operation** (a dedicated bottleneck filtering features) rather than external supervision. This creates a strong inductive bias that standard backbones lack.

- **Architecture & Efficiency:** We provided detailed specifications of the gate network and further explained why the gate mechanism can effectively learn a better shared feature space (Section 3). It is highly efficient, adding only **1.5% training time** and **negligible inference cost** (Table 8,9), ensuring it is a practical, plug-and-play module for the community.

- **Universality:** The mechanism consistently improves state-of-the-art baselines (ECB, CDAC, MME) across standard benchmarks (Table 2 & 3), proving that this "marginal" theoretical insight translates into robust, low-cost performance gains.


In conclusion, we believe this work offers a critical step forward by transforming the "common sense" of shared spaces into a **quantifiable theoretical objective** and a statistically significant architectural solution. We have addressed the reviewers' concerns with additional clarification, and have incorporated all aforementioned experiments and clarifications into the revised manuscript.

---

### Note · Authors · 2026-01-01

I have read and agree with the venue's withdrawal policy on behalf of myself and my co-authors.